# Multiphoton FLIM Analyses of Native and UVA-Modified Synthetic Melanins

**DOI:** 10.3390/ijms24054517

**Published:** 2023-02-24

**Authors:** Ana-Maria Pena, Shosuke Ito, Thomas Bornschlögl, Sébastien Brizion, Kazumasa Wakamatsu, Sandra Del Bino

**Affiliations:** 1L’Oréal Research and Innovation, 93601 Aulnay-sous-Bois, France; 2Institute for Melanin Chemistry, Fujita Health University, Toyoake 470-1192, Japan

**Keywords:** eumelanin, DHICA, DHI, Dopa, pheomelanin, Dopa-Cys, BZ-AA pure benzothiazine, DHBTCA pure benzothiazole, mixed eu-/pheo-melanins, UVA exposure, multiphoton FLIM imaging, phasor and bi-exponential fitting analyses, HPLC chemical analysis

## Abstract

To better understand the impact of solar light exposure on human skin, the chemical characterization of native melanins and their structural photo-modifications is of central interest. As the methods used today are invasive, we investigated the possibility of using multiphoton fluorescence lifetime (FLIM) imaging, along with phasor and bi-exponential fitting analyses, as a non-invasive alternative method for the chemical analysis of native and UVA-exposed melanins. We demonstrated that multiphoton FLIM allows the discrimination between native DHI, DHICA, Dopa eumelanins, pheomelanin, and mixed eu-/pheo-melanin polymers. We exposed melanin samples to high UVA doses to maximize their structural modifications. The UVA-induced oxidative, photo-degradation, and crosslinking changes were evidenced via an increase in fluorescence lifetimes along with a decrease in their relative contributions. Moreover, we introduced a new phasor parameter of a relative fraction of a UVA-modified species and provided evidence for its sensitivity in assessing the UVA effects. Globally, the fluorescence lifetime properties were modulated in a melanin-dependent and UVA dose-dependent manner, with the strongest modifications being observed for DHICA eumelanin and the weakest for pheomelanin. Multiphoton FLIM phasor and bi-exponential analyses hold promising perspectives for in vivo human skin mixed melanins characterization under UVA or other sunlight exposure conditions.

## 1. Introduction

Melanin pigment characterization is crucial for understanding human skin and hair constitutive pigmentations and how their quantity and quality are modulated in physiological conditions, in different disorders, with sunlight exposure or topical active ingredients application. A still challenging topic, especially in photo-protection, is the in vivo characterization of melanin photo-modifications (oxidative, photo-degradation, and crosslinking) and their link with the immediate and delayed pigmentations, i.e., immediate pigment darkening (IPD) and persistent pigment darkening (PPD).

IPD is a transient and reversible greyish skin pigmentation which develops during and after UV exposure and fades away a few minutes to two hours after exposure. When the skin is exposed to a sufficient dose of UVA (>10 J/cm^2^ UVA 320–400 nm), the IPD is more intense and is followed by PPD, the latter having a brown color that lasts 24 h and blends into delayed tanning 3 to 5 days after exposure [1,2,3,4]. In contrast to delayed tanning due to neomelanization, IPD and PPD are thought to be due to photo-oxidation and/or the polymerization of pre-existing melanins (IPD) or their precursors and metabolites [5,6,7,8,9].

However, the UVA-induced structural modifications of human skin melanins have not yet been characterized in vivo or ex vivo. Currently, the only method allowing the analysis of the native or photo-induced modifications of eumelanins and pheomelanins is high-performance liquid chromatography (HPLC) analysis of the melanin degradation products obtained after alkaline hydrogen peroxide oxidation (AHPO) or after reductive hydrolysis with HI [10,11]. Using this method, UVA or blue light exposure has been shown to induce the partial photo-oxidative degradation/modification of synthetic melanins and natural melanins from retinal pigmented epithelium (RPE), cultured melanocytes, and human hair samples [12,13,14,15,16]. Although very specific and allowing the discrimination between eumelanin and pheomelanin, the HPLC method requires sample collection (i.e., biopsies) and degradation and provides no information on eu-/pheo-melanin 3D distribution within the tissue. Moreover, benzothiazine pheomelanin is analyzed in a different sample aliquot (subjected to reductive hydrolysis with HI) to the one used for benzothiazole and eumelanin analysis (subjected to AHPO).

In this study, taking advantage of melanin endogenous fluorescence, we investigate the possibility of using multiphoton fluorescence lifetime imaging microscopy (FLIM) [17,18] as a non-invasive alternative method for the chemical analysis of native and UVA-exposed melanins. The fluorescence lifetime (τ) characterizes the average time spent by a fluorescent molecule in the excited state before emitting a fluorescence photon and returning to its ground state [19]. As the fluorescence lifetime, which is mostly independent of the fluorophore concentration, is sensitive to the local microenvironment of the molecule and to parameters such as pH, binding status, and molecular conformational changes [19], it could probably detect the structural photo-modifications in melanins.

Multiphoton FLIM imaging was already applied to the characterization of different melanin samples, such as synthetic 3,4-dihydroxyphenylalanine (Dopa) or Sepia melanins, human eye melanocytes, and hair and skin samples (e.g., [20,21,22,23,24,25,26]). The in vivo applicability of multiphoton and FLIM technologies, for example, in human skin clinical trials is well documented [18,24,26,27,28,29,30,31,32,33,34,35,36,37,38,39,40,41,42,43]. However, the fluorescence lifetime properties of the different types of melanins (DHI, DHICA eumelanins, benzothiazole/benzothiazine pheomelanins and mixed eu-/pheo-melanins) typically found in human skin and hair samples [44,45,46] have not yet been characterized, nor have their changes with UVA light exposure. Prior to exploring melanin photo-modifications in vivo in a complex environment with mixed melanins and various endogenous skin fluorophores, one needs to characterize these phenomena in model samples. Here, using HPLC chemical analysis and multiphoton FLIM bi-exponential fitting and phasor analyses, we characterize in tubo different types of synthetic model eumelanins, pheomelanins, and mixed eu-/pheo-melanins in their native, heat-induced, crosslinked, and UVA-modified states, with the UVA-exposed samples being a mixture of native, photo-degraded, crosslinked, and oxidized melanins. We chose to investigate the effects of UVA light following 24 h exposure at a radiance of 3.5 mW/cm^2^, which is similar to the solar radiance in Greece in June [47], and to a longer 7-day exposure period, which was chosen in order to maximize the UVA-induced modifications in the melanins.

## 2. Results

We characterized the endogenous fluorescence lifetime properties of native and UVA-exposed synthetic melanins by multiphoton FLIM imaging and, using a portion of the same samples, we also analyzed them by HPLC chemical analysis to confirm the presence of UVA- and heat-induced structural changes. We studied (i) native melanins; (ii) melanins exposed to UVA light at a radiance of 3.5 mW/cm^2^ for 1 and 7 days to induce different levels of melanin photo-degradation, C-C crosslinking, and some decarboxylation; and (iii) melanins exposed for 8 days at a temperature of 100 °C to induce melanin crosslinking and extensive decarboxylation [48]. Depending on the melanin type and exposure conditions, the samples were prepared in solution, suspension, or powder. We will first present the HPLC results before addressing the multiphoton FLIM results. The moderate, strong, and very strong modulations indicated in the text are based upon the calculation of the effect size statistical parameter.

### 2.1. UVA Effects in Melanin Solutions Confirmed by HPLC Chemical Analysis

The modifications induced in the melanin solutions following 1 day and 7 days of UVA exposure at 3.5 mW/cm^2^ are indicated in Table 1 and, for some parameters, represented in Figure 1. Notably, UVA exposure conditions are not physiological but aim at maximizing the degree of melanin photo-degradation. General information on the different types of melanins and their characterization by HPLC is given in Section 4.1.

For 5,6-dihydroxyindole-2-carboxylic acid (DHICA) eumelanin, a progressive decrease in pyrrole-2,3,5-tricarboxylic acid (PTCA) (Figure 1a) was observed after 1 day (1.41× ↓) and 7 days (1.60× ↓) of UVA exposure. As indicated by the changes in the free PTCA (Figure 1b) and pyrrole-2,3,4,5-tetracarboxylic acid (PTeCA) (Figure 1c) parameters, the native DHICA eumelanin was partially converted to oxidized (3.20× ↑ in free PTCA after 1 day and 7.12× ↑ after 7 days) and crosslinked (2.08× ↑ in PTeCA after 7 days) eumelanins. The eumelanin oxidation changes were assessed via the free/total PTCA ratio (Figure 1d), which showed, in our conditions, an increase after 1 day (4.50× ↑) and 7 days (11.38× ↑) of UVA exposure.

Changes in the eumelanin crosslinking parameter PTeCA (Figure 1c) with UVA exposure were also observed for 5,6-dihydroxyindole (DHI) eumelanin (1.16× ↑ after 1 day; 1.40× ↑ after 7 days) and for Dopa eumelanin (1.26× ↓ after 1 day; 1.11× ↑ after 7 days).

UVA-induced modifications were also detected in the samples containing pheomelanin, as indicated by the changes in the 4-amino-3-hydroxyphenylalanine (4-AHP) (Figure 1e) and thiazole-2,4,5-tricarboxylic acid (TTCA) (Figure 1f) parameters. The native pheomelanin (Pheo - Dopa-Cysteine (Cys)-1-1) and mixed eu-/pheo-melanin (Eu/Pheo - Dopa-Cys-4-1 at a ratio of 75/25) solutions were a mixture of both benzothiazine and benzothiazole (see Table 1). The ratio of benzothiazine to benzothiazole pheomelanins was 17.9 in the native pheomelanin and 10.21 in the mixed eu-/pheo-melanin solutions. We detected 2.54× more benzothiazine pheomelanin and 1.54× more benzothiazole pheomelanin in the native pheomelanin than in the mixed eu-/pheo-melanin solutions.

After UVA exposure, benzothiazine pheomelanin is converted into benzothiazole pheomelanin.

For pheomelanin (Pheo - Dopa-Cys-1-1), a progressive decrease in 4-AHP benzothiazine pheomelanin (Figure 1e) was observed after 1 day (1.60×) and 7 days (3.38×) of UVA exposure. As indicated by the changes in TTCA (Figure 1f) and thiazole-4,5-dicarboxylic acid (TDCA) (Table 1), the benzothiazole pheomelanin increased, respectively, by the factors of 1.16 and 1.86. The decrease in benzothiazine pheomelanin is higher than the increase in benzothiazole pheomelanin. The lower response of the TTCA parameter could be explained by the further TTCA photo-oxidation to unidentified products. Moreover, some of the benzothiazine is oxidized to benzothiazole pheomelanin, affording TDCA. The changes due to pheomelanin oxidation are expressed as the ratio of TTCA/4-AHP (Figure 1g), which shows an increase of 1.86× after 1 day and 6.30× after 7 days of UVA exposure.

For mixed eu-/pheo-melanin (Eu/Pheo -Dopa-Cys-4-1 at a ratio of 75/25), after 7 days of exposure (no measurement data at 1 day), we measured a 2.44× decrease in 4-AHP benzothiazine pheomelanin (Figure 1e), along with a 1.32× and a 1.47× increase, respectively, in TDCA and TTCA benzothiazole pheomelanins (Figure 1f). The corresponding TTCA/4-AHP pheomelanin oxidation ratio (Figure 1g) increased by a factor which was 3.58× smaller compared to that of the pheomelanin solution. The reasons for the lower response (smaller increase in TDCA and TTCA benzothiazole pheomelanins compared to the higher decrease in 4-AHP benzothiazine pheomelanin) to UVA exposure in mixed eu-/pheo-melanin could be the same as those for Pheo - Dopa-Cys-1-1, as discussed above.

Mixed eu-/pheo-melanin also contains DHICA and DHI eumelanins (1.36× more PTCA - DHICA eumelanin than PDCA - DHI eumelanin) (Table 1) After UVA exposure, we observed a decrease in PTCA (1.69× ↓) (Figure 1a) and PDCA (1.31× ↓) (Table 1) and almost no change in the free PTCA eumelanins oxidation (1.00×) (Figure 1b) and PTeCA crosslinking (1.03× ↑) (Figure 1c) parameters. The eumelanin oxidation free/total PTCA ratio increase (1.27× ↑) (Figure 1d) with UVA exposure is due to the decrease in PTCA. Notably, the synthetic mixed eu-/pheo-melanin solution cannot be prepared with enough quantities of DHICA units to detect the modulations related to eumelanin’s oxidation and crosslinking.

Altogether, these data suggest that in mixed eu-/pheo-melanins at a 75/25 ratio pheomelanin, despite its small amount compared to the eumelanin, undergoes more changes than eumelanin upon 7 days of UVA exposure.

### 2.2. Heat-Induced Crosslinking Effects in Melanin Powders Evidenced by HPLC Chemical Analysis

Melanin powder samples were studied in their native and heat-induced crosslinking state. After heating at 100 °C for 8 days (Table 1), we observed an increase in the PTeCA eumelanin crosslinking parameter for DHICA (4.52× ↑), DHI (1.98× ↑), Dopa (1.62× ↑), and mixed DHI + DHICA (1.99× ↑) eumelanins. The PTeCA parameter showed almost no change in pheomelanin (Pheo - Dopa-Cys-1-1) (1.18× ↑) and mixed eu-/pheo-melanin (Eu/Pheo - Dopa-Cys-2-1 at a ratio of 50/50) (1.29× ↑).

For the DHICA-containing samples, a decrease in the PTCA (DHICA eumelanin) parameter was observed after heating (DHICA 10.70× ↓, mixed DHI + DHICA 4.85× ↓), thus confirming the loss of the carboxyl group of DHICA due to decarboxylation [48,49]. Notably, PTCA arises not only from DHICA moiety but also from the DHI moiety attached at the C2 position to the adjacent unit. These modifications can be summarized by the combined PTeCA/PTCA eumelanin crosslinking ratio (also shown to be highly modulated with UVA exposure in the DHICA eumelanin-containing samples): DHICA (48.34× ↑), DHI (1.61× ↑), Dopa (1.17× ↑), mixed DHI + DHICA (9.65× ↑), Pheo - Dopa-Cys-1-1 (1.90× ↑), and mixed Eu/Pheo - Dopa-Cys-2-1 (2.19× ↑).

The PDCA (DHI eumelanin) parameter showed no changes upon heating except for those of Eu-DHICA (1.69× ↓) and Eu/Pheo - Dopa-Cys-2-1 (1.31× ↓).

The exposure of melanin samples to heating induced not only eumelanin crosslinking and decarboxylation but also changes in the pheomelanin oxidation ratio (TTCA/4-AHP) that increased 3.54× in the pheomelanin (Pheo - Dopa-Cys-1-1) and 5.39× in the mixed eu-/pheo-melanin (Eu/Pheo - Dopa-Cys-2-1 at a ratio of 50/50). This modification reflects the partial conversion of 4-AHP benzothiazine pheomelanin (1.91× ↓) into TTCA benzothiazole pheomelanin (1.85× ↑), which appears to have been induced simultaneously with the eumelanin crosslinking in the mixed eu-/pheo-melanin.

### 2.3. Acquiring and Analyzing Multiphoton FLIM Images on Different Melanin Samples

We performed time-domain two-photon excited fluorescence (2PEF) lifetime measurements upon excitation at 760 nm and detection in the 409–680 nm range (see Section 4.4). We processed the 2PEF intensity decays in every pixel of the images, with both phasor analysis, based on fast Fourier transform [26,50,51,52], and bi-exponential fitting analysis [20,21,22,24,26].

For the phasor analysis (see Section 4.6), we first calculated the phase $\tau_{\varphi }$ and modulation $\tau_{m}$ lifetimes and the phasor plots (scatter of phasor *s* versus *g* parameters), color coded by these lifetime parameters. These phasors plots are FLIM fingerprints of the different native and UVA-exposed melanins, allowing the regrouping of the melanin pixels with similar fluorescence lifetime properties and the visualization in one shot of the samples’ differences.

Given that UVA-exposed melanins are a mixture of native melanins and UVA-modified (oxidized, crosslinked, and photo-degraded) melanins, we considered implementing the phasor analysis to calculate the fraction of melanin species structurally modified by the UVA exposure and contributing to the global fluorescence signal. Using native melanins as reference samples, we quantified the relative concentration/fraction of UVA-modified melanins, $f_{UVA\;Mel}$, by graphically measuring the distance of each experimental UVA-exposed melanin pixel in the phasor plot to the average location of the native melanins.

For the bi-exponential analysis (see Section 4.5), we calculated the $\tau_{1}$ short and $\tau_{2}$ long fluorescence lifetimes, their relative contributions, $a_{1}\left[\%\right]$, $a_{2}\left[\%\right]$, as well as their combination parameters, the $\tau_{AvInt}$ intensity- and $\tau_{AvAmp}$ amplitude-weighted average lifetimes. The intensity- and amplitude-weighted average lifetimes can be used to assess the global modifications of the multiphoton fluorescence decay pattern, but to interpret these parameters, one has to individually assess the $\tau_{1}$, $\tau_{2}$, $a_{1}\left[\%\right]$, and $a_{2}\left[\%\right]$ parameters.

### 2.4. Multiphoton FLIM Imaging Discriminates Native Eumelanins, Pheomelanin, and Mixed eu-/pheo-melanins

Representative raw two-photon excited fluorescence intensity decays of native melanins (no UVA exposure) in solution or suspension are shown in Figure 2.

In our experimental conditions, we observed differences in the signal intensities between the native melanins (see intensity decays in Figure 2a and mean 2PEF image intensities in Figure 3), with the lowest signals being detected for DHI eumelanin.

As shown in Figure 2b and confirmed by both the phasor analysis (Figure 3, showing a mixed fluorescence species fingerprint, and Figure 4) and the bi-exponential fitting analysis (Figure 5 and Figure 6), the fluorescence signal of the studied melanins exhibits a bi-exponential decay (Figure 2a,b) with a main relative contribution $a_{1}\left[\%\right]$ of the $\tau_{1}$ short fluorescence lifetime to the global fluorescence signal (Figure 5 and Figure 6 and Table 2).

Native DHI eumelanin presents a phasor plot with a comet-like fingerprint pointing towards very short lifetime contributions (Figure 3) and a broad phasor distribution (broad *s* and *g* values—Figure 4a,b, Table 2), with pixels ranging from longer phase and modulation lifetimes to shorter ones (average $\tau_{\varphi }$ = 0.55 ± 0.22 ns and $\tau_{m}$ = 2.61 ± 0.68 ns; Figure 4c,d; Table 2). Its mean short and long fluorescence lifetime components also indicate a broader distribution ($\tau_{1}$ = 118 ± 52 ps; $a_{1}\left[\%\right]$ = 91.78 ± 6.26%; $\tau_{2}$ = 1.92 ± 0.29 ns; Figure 5 and Figure 6a–d, Table 2). As observed in the 2PEF intensity images (Figure 3), the native DHI eumelanin suspension contains some aggregates characterized by a higher 2PEF intensity and smaller lifetimes. The presence of both structures of the homogenous suspensions of the DHI polymers and DHI aggregates accounts for this heterogenous phasor plot fingerprint.

Native DHICA eumelanin is characterized by a more homogenous phasor plot distribution (Figure 3) with the highest *g* and the smallest *s* values (Figure 4a,b, Table 2) corresponding to very small phase lifetimes (average $\tau_{\varphi }$ = 0.19 ± 0.05 ns and $\tau_{m}$ = 1.00 ± 0.17 ns; Figure 4c,d; Table 2) The FLIM bi-exponential analysis (Figure 5 and Figure 6, Table 2) indicates that its fluorescence signal is mainly dominated by the short fluorescence lifetime species (smallest $\tau_{1}$ = 85 ± 2 ps; highest $a_{1}\left[\%\right]$ = 98.43 ± 0.27%; $\tau_{2}$ = 2.1 ± 0.1 ns; Figure 5 and Figure 6a–d, Table 2).

These two types of eumelanin, which are quite similar in terms of molecular structure (DHI and DHICA), show very strong differences for all the phasor parameters (Figure 4a–d) and variable differences in the bi-exponential analysis parameters: moderate differences in the $\tau_{1}$ short fluorescence lifetime, strong differences in the $a_{1}\left[\%\right]$ and $a_{2}\left[\%\right]$ relative contributions, and very strong differences in the $\tau_{2}$ long fluorescence lifetime (Figure 6a–d). Moderate to strong differences were also evidenced by the combined $\tau_{Av\;Int}$ intensity- and $\tau_{Av\;Amp}$ amplitude-weighted average lifetimes (Figure 6e,f) (varying from $\tau_{Av\;Int}=$ 1.09 ± 0.37 ns; $\tau_{Av\;Amp}$ = 0.27 ± 0.17 ns for DHI eumelanin to $\tau_{Av\;Int}=$0.65 ± 0.09 ns; $\tau_{Av\;Amp}$ = 0.12 ± 0.01 ns for DHICA eumelanin; Table 2). These differences in the DHI and DHICA eumelanins are probably due to the presence of carboxylic acid in DHICA eumelanin.

We also investigated native Dopa eumelanin, which is often used in the literature as a standard for eumelanin. We found very strong differences between the Dopa eumelanin and both of the DHICA and DHI eumelanins for all the phasor parameters ($\tau_{\varphi }$ = 0.51 ± 0.10 ns and $\tau_{m}$ = 1.26 ± 0.18 ns; Figure 4a–d; Table 2). With bi-exponential fitting ($\tau_{1}$ = 129 ± 12 ps; $a_{1}\left[\%\right]$ = 93.55 ± 1.47%; $\tau_{2}$ = 1.89 ± 0.1 ns; $\tau_{Av\;Int}=$1.01 ± 0.12 ns; $\tau_{Av\;Amp}$ = 0.24 ± 0.04 ns Figure 5 and Figure 6a–f, Table 2) the differences were also very strong between the Dopa and DHICA eumelanins for all the parameters and only moderate between the Dopa and DHI eumelanins using the $\tau_{1}$ parameter. This clearly highlights the advantage of phasor over bi-exponential fitting in discriminating melanins. Moreover, as for DHI eumelanin, Dopa eumelanin is also a suspension with high 2PEF intensity aggregates characterized by small phase lifetimes (Figure 3). Altogether, the presence of both structures of the homogenous suspension of the Dopa polymers and Dopa aggregates accounts for its heterogenous phasor plot fingerprint.

Native pheomelanin (Pheo - Dopa-Cys-1-1), a mixture of benzothiazine and benzothiazole quantified by HPLC at a 17.9 ratio of 4-AHP/TTCA (Table 1), shows a homogenous phasor fingerprint (Figure 3) with the following characteristics for the phasor ($\tau_{\varphi }$ = 0.61 ± 0.06 ns and $\tau_{m}$ = 1.42 ± 0.09 ns; Figure 4a–d; Table 2) and the bi-exponential fitting ($\tau_{1}$ = 126 ± 10 ps; $a_{1}\left[\%\right]$ = 92.38 ± 1.00%; $\tau_{2}$ = 2.02 ± 0.09 ns; $\tau_{Av\;Int}=$1.20 ± 0.09 ns; $\tau_{Av\;Amp}$ = 0.27 ± 0.03 ns Figure 5 and Figure 6a–f, Table 2) parameters.

Compared to DHI eumelanin, native pheomelanin shows very strong differences in the phasor *g*, *s*, and $\tau_{m}$ modulation lifetime parameters and no differences in the phase lifetime, whereas only moderate differences were evidenced with the $\tau_{2}$ and $\tau_{Av\;Int}$ bi-exponential fitting lifetimes. Compared to DHICA eumelanin, very strong differences were evidenced for all the phasor parameters and all the bi-exponential fitting parameters, except for $\tau_{2}$, which only showed moderate differences. Compared to Dopa eumelanin, native pheomelanin also showed very strong differences for all the phasor parameters and moderate ($\tau_{2},\;\;a_{1}\left[\%\right],\;\;a_{2}\left[\%\right],\;\tau_{Av\;Amp})$ to very strong ($\tau_{Av\;Int})$ differences for the bi-exponential fitting parameters.

Native mixed eu-/pheo-melanin (Eu/Pheo - Dopa-Cys-4-1 at a ratio of 75/25) is a mixture of 75% eumelanins (composed of 90% DHI and 10% DHICA) and 25% pheomelanin (quantified by HPLC at a 10.21 ratio of benzothiazine (4-AHP) to benzothiazole (TTCA); see HPLC quantification results, Table 1). This type of mixed melanin solution shows a homogenous phasor fingerprint (Figure 3) with the following characteristics for the phasor ($\tau_{\varphi }$ = 0.38 ± 0.02 ns and $\tau_{m}$ = 0.92 ± 0.06 ns; Figure 4a–d; Table 2) and the bi-exponential fitting ($\tau_{1}$ = 104 ± 0.3 ps; $a_{1}\left[\%\right]$ = 95.45 ± 0.17%; $\tau_{2}$ = 1.83 ± 0.06 ns; $\tau_{Av\;Int}=$0.89 ± 0.03 ns; $\tau_{Av\;Amp}$ = 0.18 ± 0.01 ns Figure 5 and Figure 6a–f, Table 2) parameters.

Compared to DHI eumelanin, native mixed eu-/pheo-melanin shows differences in the phasor parameters varying from strong (phasor *s* and $\tau_{\varphi }$ phase lifetime) to very strong ($\tau_{m}$) and moderate differences in the $\tau_{2}$ and $\tau_{Av\;Amp}$ bi-exponential fitting parameters. Compared to DHICA eumelanin, the differences are very strong for the phasor *s* and $\tau_{\varphi }$ phase lifetime and all the bi-exponential fitting parameters. Regarding the differences with the Dopa eumelanin, they are very strong for all the phasor parameters and all the bi-exponential fitting parameters except $a_{1}\left[\%\right],$ which shows a moderate difference. Finally, the native mixed eu-/pheo-melanin to native pheomelanin differences are all very strong for all the quantified parameters.

### 2.5. UVA Exposure Globally Induces Dose- and Melanin Type-Dependent Modulations in Multiphoton FLIM Lifetime Parameters

The changes in multiphoton FLIM parameters occurring in the melanin solutions after 1 and 7 days of UVA exposure are illustrated in Figure 2, Figure 3 and Figure 5 and their quantification is shown in Figure 4 and Figure 6 and Table 2. As demonstrated by chemical analysis, UVA exposure induces structural changes in melanins that impact the fluorescence lifetime properties. The 2PEF intensity decays in Figure 2c–e clearly show differences between the native melanins and the 1 day and 7 days of UVA exposure, respectively, with the smallest difference being observed for pheomelanin. These changes translate into a visible shift in the phasor fingerprint along with a modification of the phasor (*s*, *g*) pixel distribution (see the phasor plots in Figure 3).

For the DHICA eumelanin solution, a progressive and very strong modulation of all the phasor parameters was observed with the UVA exposure (Figure 4, Table 2). The mean values of the phase lifetime $\tau_{\varphi }$ increased from 0.19 ns (no exposure) to 0.65 ns (1-day UVA) and 1.03 ns (7-day UVA), whereas the modulation lifetime $\tau_{m}$ increased from 1.00 ns (no exposure) to 1.96 ns (1-day UVA) and 2.25 ns (7-day UVA). Very strong modulations were also evidenced by the bi-exponential fitting analysis (Figure 6, Table 2). The short fluorescence lifetime $\tau_{1}$ progressively increased from 85 ps (no exposure) to 109 ps (1-day UVA) and 210 ps (7-day UVA), along with a progressive decrease in its relative contribution $a_{1}\left[\%\right]$ from 98.4% (no exposure) to 93.8% (1-day UVA) and 80.9% (7-day UVA). The long fluorescence lifetime $\tau_{2}$ (2.10 ns, no exposure) increased only moderately after 1 day of UVA exposure (2.20 ns) and very strongly after 7 days (2.52 ns). These changes translate into very strong modulations of the combination parameters of the intensity-weighted average lifetime $\tau_{Av\;Int}$ (0.65 ns (no exposure); 1.30 ns (1-day UVA) and 1.92 ns (7-day UVA)) and the amplitude-weighted average lifetime $\tau_{Av\;Amp}$ (0.12 ns (no exposure); 0.25 ns (1-day UVA) and 0.65 ns (7-day UVA)).

According to the HPLC results, native DHICA eumelanin is partially converted by UVA light to crosslinked and oxidized eumelanins. The UVA-exposed DHICA eumelanin solution contains four types of molecules: native, oxidized, crosslinked, and photo-degraded eumelanins. Given the absence of the fluorescence signal in both the native and the heating-induced crosslinked melanin powders and the fact that totally oxidized melanin samples cannot be obtained, we cannot go further into the attribution of the UVA-induced FLIM changes to a specific compound. The fluorescence signal of UVA-exposed DHICA eumelanin is a global signal arising from the four types of molecules, but given their molecular structure, oxidized eumelanin probably has a higher impact on the fluorescence lifetime properties.

To go further into the analysis of this signal, using the native DHICA eumelanin as a reference, we implemented phasor analysis to calculate the fraction of melanin species structurally modified by the UVA exposure (Figure 6e). For that, we first quantified the average *g* and *s* coordinates of all the pixels of the native DHICA eumelanin solution (see the center of the pink circle in Figure 6e). The $f_{UVA\;Mel}$ relative fraction of the UVA-modified eumelanin species was calculated by graphically measuring the distance (e.g., the green and orange brackets in Figure 6e) of every UVA-exposed experimental data point to the reference native eumelanin average *g* and *s* position. Figure 6e illustrates the $f_{UVA\;Mel}$ relative fraction calculation for eumelanin, but we implemented the same process for all the other melanin types. The quantification results of $f_{UVA\;Mel}$ are given in Figure 6f.

Upon UVA exposure, the $f_{UVA\;Mel}$ in the DHICA eumelanin progressively and very strongly increased with the UVA dose (Figure 6f, Table 2). Its mean values increased from 3.2 ± 1.9% (no exposure) to 25.2 ± 1.7% (1-day UVA) and 37.1 ± 8.0% (7-day UVA). Compared to the other melanin types, DHICA eumelanin exhibited the highest modulation in the relative fraction of the UVA-modified species, followed by mixed eu-/pheo-melanins, DHI eumelanin, Dopa eumelanin, and pheomelanin.

Although quite similar in terms of molecular structure to DHICA eumelanin, DHI eumelanin’s fluorescence lifetime response to UVA exposure is different. Concerning the phasor parameters (Figure 6f, Table 2), the mean values of the phase lifetime $\tau_{\varphi }$ progressively and very strongly increased from 0.55 ns (no exposure) to 1.17 ns (1-day UVA) and 1.27 ns (7-day UVA), whereas the modulation lifetime $\tau_{m}$ increased very strongly only after 7 days of exposure (2.61 ns (no exposure) to 2.84 ns (1-day UVA) and 3.96 ns (7-day UVA)). The phasor *g* parameter showed a strong and very strong decrease, respectively, at 1 day and 7 days of UVA exposure, while the phasor *s* parameter showed a very strong increase after 1 day of UVA exposure and was followed by a decrease after 7 days (with a higher value than the no exposure condition).

As indicated by the HPLC results, upon UVA exposure native DHI eumelanin was partially (1.35× decrease in PDCA after 7 days) converted to crosslinked eumelanin (1.4× increase in PTeCA) and oxidized eumelanin (5× increase in free PTCA, characterizing C2-bridged DHI eumelanin peroxidation). Once more, the multiphoton FLIM signal of UVA-exposed DHI eumelanin is a mixture of fluorescence photons emitted by the different native, crosslinked, photo-degraded, and oxidized compounds. The phasor FLIM analysis evidenced a progressive and very strong increase in $f_{UVA\;Mel}$, the relative fraction of the UVA-modified DHI eumelanin species, although to a lesser extent compared to DHICA eumelanin (10.2 ± 7.0% (no exposure) to 17.0 ± 2.8% (1-day UVA) and 23.1 ± 4.1% (7-day UVA)).

Strong and very strong modulations were also evidenced by the bi-exponential fitting analysis (Figure 6, Table 2).

The short fluorescence lifetime $\tau_{1}$ very strongly increased from 118 ps (no exposure) to 304 ps (1-day UVA), and followed a strong decrease at 278 ps (7-day UVA). Its relative contribution $a_{1}\left[\%\right]$ very strongly decreased from 91.8% (no exposure) to 74.3% (1-day UVA) and remained constant at 74.1% (7-day UVA). The long fluorescence lifetime $\tau_{2}$ (1.92 ns no exposure) very strongly increased after 1 day (2.42 ns) and 7 days (2.57 ns) of UVA exposure, respectively. These changes translate into very strong modulations of the intensity-weighted average lifetime $\tau_{Av\;Int}$ at all the time points (1.09 ns (no exposure); 1.85 ns (1-day UVA) and 2.02 ns (7-day UVA)) and of the amplitude-weighted average lifetime $\tau_{Av\;Amp}$ after 1 day of exposure (0.27 ns (no exposure); 0.85 ns (1-day UVA) and 0.86 ns (7-day UVA)).

The changes in Dopa eumelanin upon UVA exposure could also be evidenced with multiphoton FLIM imaging. According to the phasor analysis (Figure 4, Table 2), the mean values of the phase lifetime $\tau_{\varphi }$ very strongly increased from 0.57 ns (no exposure) to 0.93 ns (1-day UVA) and decreased to 0.67 ns (7-day UVA), whereas the modulation lifetime $\tau_{m}$ very strongly increased after 1 day of UVA exposure and remained constant (1.26 ns (no exposure); 1.86 ns (1-day UVA); and 1.85 ns (7-day UVA)). The phasor *g* parameter showed a very strong decrease and increase (with a lower value than the no exposure condition), respectively, while the phasor *s* parameter varied in the opposite manner, showing a very strong increase and decrease (higher value than the no exposure condition) after 1 day and 7 days of UVA exposure, respectively.

Using HPLC analysis, although we cannot estimate the changes in the native Dopa eumelanin upon UVA exposure (there is no parameter to assess them), we observed an increase in the PTeCA eumelanin crosslinking after 1 day (1.3×) and 7 days (1.1×), along with an increase in oxidized eumelanin after 7 days (3×). The phasor analysis evidenced a very strong increase in $f_{UVA\;Mel}$, the relative fraction of the UVA-modified species, after 1 day, followed by a decrease at 7 days (with a higher value than the no exposure condition), both values being smaller compared to the DHICA and DHI eumelanins: 4.4 ± 3% (no exposure); 18.6 ± 3.2% (1-day UVA); and 13.6 ± 5.5% (7-day UVA).

The bi-exponential fitting analysis (Figure 6, Table 2) also highlights a very strong increase followed by a decrease in the $\tau_{1}$ (129 ps (no exposure); 228 ps (1-day UVA); and 155 ps (7-day UVA)), $a_{1}\left[\%\right]$ (93.6% (no exposure); 82.9% (1-day UVA); and 91.8% (7-day UVA)), $\tau_{Av\;Int}$ (1.01 ns (no exposure); 1.60 ns (1-day UVA); and 1.33 ns (7-day UVA)), and $\tau_{Av\;Amp}$ (0.24 ns (no exposure); 0.60 ns (1-day UVA); and 0.33 ns (7-day UVA)) parameters at all the time points and a very strong increase in $\tau_{2}$ only after 1 day (1.89 ns (no exposure); 2.27 ns (1-day UVA); and 2.27 ns (7-day UVA)).

Pheomelanin (Pheo - Dopa-Cys-1-1, a 17.9 ratio of benzothiazine (4-AHP) to benzothiazole (TTCA)) showed the slightest modulations in both types of FLIM parameters with moderate, strong, and very strong differences depending on the parameters.

According to the phasor analysis (Figure 4, Table 2), the phase lifetime $\tau_{\varphi }$ (0.61 ns (no exposure); 0.72 ns (1-day UVA); and 0.61 ns (7-day UVA)) and the modulation lifetime $\tau_{m}$ (1.42 ns (no exposure); 1.60 ns (1-day UVA); and 1.47 ns (7-day UVA)) very strongly increased only after 1 day of UVA exposure. The phasor *g* and *s* parameter showed a very strong decrease and a strong increase, respectively, after 1 day of UVA exposure.

The HPLC results indicate that pheomelanin was oxidized by UVA light, i.e., benzothiazine was partially converted to benzothiazole, as evidenced by the pheomelanin oxidation ratio (TTCA benzothiazole/4-AHP benzothiazine ratio increase of 1.87× and 6.34× after 1 and 7 days of UVA exposure, respectively). The changes in the *g* and *s* parameters translate into a slight modulation of the $f_{UVA\;Mel}$ relative fraction of the UVA-modified species: strong increase at 1 day and moderate decrease after 7 days (2.5 ± 1.3% (no exposure); 5.3 ± 2.2% (1-day UVA); and 1.5 ± 0.8% (7-day UVA).

The bi-exponential fitting results (Figure 6, Table 2) also revealed small changes in the pheomelanin fluorescence lifetime properties. There was no change in the short fluorescence lifetime $\tau_{1}$ after 1 day, but there was a strong decrease after 7 days (126 ps (no exposure); 132 ps (1-day UVA); and 113 ps (7-day UVA)). Its relative contribution $a_{1}\left[\%\right]$ strongly decreased at 1 day (92.4% (no exposure); 90.6% (1-day UVA); and 92.5% (7-day UVA)). The long fluorescence lifetime $\tau_{2}$ moderately increased and decreased (2.02 ns (no exposure); 2.08 ns (1-day UVA); and 1.95 ns (7-day UVA)) after 1 day and 7 days of UVA exposure, respectively. These changes translate into a strong increase at 1 day in both of the combined lifetime parameters, $\tau_{Av\;Int}$ (1.20 ns (no exposure); 1.34 ns (1-day UVA); and 1.19 ns (7-day UVA)) and $\tau_{Av\;Amp}$ (0.27 ns (no exposure); 0.32 ns (1-day UVA); and 0.25 ns (7-day UVA)).

The absence or the slight UVA-induced modifications in the fluorescence lifetime parameters of pheomelanin suggest that the two types of pheomelanins probably have similar fluorescence lifetime characteristics. The analysis of the benzothiazine and benzothiazole monomers (see Section 2.7) strengthens this hypothesis. Thus, we can extrapolate that the two types of pheomelanin polymers have similar multiphoton FLIM properties in the investigated solution and one cannot use this method to assess their UVA-induced modifications.

The mixed eu-/pheo-melanin (Eu/Pheo - Dopa-Cys-4-1 at a ratio of 75/25) was only investigated before and after 7 days of UVA exposure. All the multiphoton FLIM parameters were modulated with the UVA exposure. After 7 days of UVA exposure, all the phasor parameters (Figure 4, Table 2) were very strongly modulated: very strong increase in both the phase $\tau_{\varphi }$ (0.38 ns (no exposure) and 0.98 ns (7-day UVA)) and the modulation $\tau_{m}$ (0.92 ns (no exposure) and 1.93 ns (7-day UVA)) lifetimes, while the phasor *g* and *s* parameters very strongly decreased and increased, respectively. A very strong modulation was also quantified for the $f_{UVA\;Mel}$ relative fraction of the UVA-modified species, which increased from 1.1 ± 0.6% (no exposure) to 28.8 ± 1.4% (7-day UVA).

The FLIM bi-exponential fitting parameters (Figure 6, Table 2) were all very strongly modulated. The short fluorescence lifetime $\tau_{1}$ (104 ps (no exposure) and 174 ps (7-day UVA)) and the long fluorescence lifetime $\tau_{2}$ (1.83 ns (no exposure) and 2.25 ns (7-day UVA)) increased very strongly. The relative contribution $a_{1}\left[\%\right]$ very strongly decreased (95.4% (no exposure) and 82.0% (7-day UVA)). These changes are reflected in a very strong increase at 7 days of UVA exposure in both the combined lifetime parameters, $\tau_{Av\;Int}$ (0.89 ns (no exposure) and 1.71 ns (7-day UVA)) and $\tau_{Av\;Amp}$ (0.18 ns (no exposure) and 0.55 ns (7-day UVA)).

In this type of mixed melanins, after 7 days of UVA exposure, we evidenced, using HPLC analysis (see Section 2.1), a higher decrease factor in the benzothiazine compared to the increase factor in the benzothiazole and no change or very slight modifications in the DHI, DHICA eumelanins, crosslinking, and oxidation. Altogether, the HPLC data suggest that pheomelanin, despite its small amount compared to eumelanin in the mixed 75/25 ratio eu-/pheo-melanin, undergoes more changes than DHICA eumelanin upon 7 days of UVA exposure. Notably, the synthetic mixed eu-/pheo-melanin solution cannot be prepared with enough quantities of DHICA units for it to be possible to detect the modulations related to eumelanin’s oxidation and crosslinking by the HPLC method.

Conversely, the multiphoton FLIM parameters of mixed eu-/pheo-melanin were strongly modulated upon 7 days of UVA exposure. Considering that the multiphoton FLIM signal of the mixed benzothiazine and the benzothiazole pheomelanins was not modified by UVA exposure, we can assume that these changes in the mixed eu-/pheo-melanin solution are probably due to UVA-induced modifications in the DHI and DHICA eumelanins.

### 2.6. Multiphoton FLIM Analysis of Heat-Induced Crosslinking Effects in Melanin Powders

To identify the influence of the crosslinking effects on the fluorescence lifetime change evidenced upon UVA exposure, we studied melanin powders in either native or heating-induced crosslinking. Globally, all the samples did not have fluorescence, but sometimes, a small intensity signal with a very fast decay equivalent to an instrumental response function (IRF) signal was detected. Therefore, the contribution of these crosslinking effects remains unknown.

### 2.7. Multiphoton FLIM Characterization of Native Eumelanin and Pheomelanin Monomers

Given the absent or the slightly UVA-induced modifications in the fluorescence lifetime parameters of pheomelanin, we thought that benzothiazine and benzothiazole pheomelanins probably had similar fluorescence lifetime properties. “Pure” native benzothiazine pheomelanin and “pure” native benzothiazole pheomelanin polymers cannot be synthesized to verify this hypothesis. Therefore, we decided to analyze the monomers of benzothiazine and benzothiazole as well as the monomers of DHI and DHICA eumelanin.

We found that the native benzothiazine and benzothiazole pheomelanin monomers presented quite similar fluorescence lifetime properties. Their mean short and long fluorescence lifetimes were the same (0.35 ns and ~2.2 ns), but their relative contribution was different (35% for benzothiazine versus 72% for benzothiazole). The difference in the relative contribution is reflected in the intensity- and amplitude-weighted average lifetimes, which were, respectively, around 2.07 ns and 1.73 ns for benzothiazine and 1.58 ns and 0.89 ns for the benzothiazole pheomelanin monomers. The pheomelanin monomers had an approximately 2.8× longer $\tau_{1}$ fluorescence lifetime compared to the native pheomelanin in solution (a mixture of benzothiazine and benzothiazole). Once again, we think that the packing of pheomelanin molecules is responsible for this different behavior between the monomers and the polymers.

Similarly, the native DHI and DHICA eumelanin monomers presented different fluorescence lifetime properties. The short fluorescence lifetime $\tau_{1}$ was found to be, on average, around 270 ps for the DHI monomer and 182 ps for the DHICA monomer, approximately 2-3× higher than that for the native DHI and DHICA eumelanins in solution. A small difference was also evidenced in their mean $\tau_{2}$ long fluorescence lifetime (1.34 ns for DHI versus 1.26 ns for DHICA). For the DHI eumelanin monomers, we found a relative contribution $a_{1}\left[\%\right]$ of 69%, whereas for DHICA (93%) it was comparable to the DHI, DHICA, and Dopa eumelanins in solution. These differences are also evidenced by the intensity-weighted average lifetime $\tau_{Av\;Int}$ (1.00 ns for DHI versus 0.54 ns for DHICA) and the amplitude-weighted average lifetime $\tau_{Av\;Amp}$ (0.60 ns for DHI versus 0.25 ns for DHICA). Once again, the differences observed between the eumelanin monomers in powder and the eumelanin polymers in solution could be due to the differences in their molecular organization.

## 3. Discussion

In this study, we took advantage of the natural endogenous fluorescence lifetime properties of melanins and investigated the possibility of using multiphoton FLIM imaging along with both phasor and bi-exponential fitting analyses to differentiate native eumelanins, pheomelanins, and mixed eu-/pheo-melanins, as well as to detect their structural modifications induced by heating (crosslinked melanins) and UVA exposure (a mixture of native, photo-degraded, oxidized, and crosslinked melanins).

The melanin samples were first characterized by the HPLC chemical analysis of the melanin degradation products, which provided the evidence for the following findings:The heating of the melanin powders led to crosslinking (increase in PTeCA) and decarboxylation (decrease in PTCA) in the eumelanins and the partial conversion of benzothiazine pheomelanin to benzothiazole pheomelanin (decrease in 4-AHP and increases in TTCA and TDCA);The UVA exposure of melanin solutions and suspensions led to the crosslinking and peroxidative degradation (increase in PTeCA and free PTCA) in the eumelanins and the peroxidative conversion of benzothiazine pheomelanin to benzothiazole pheomelanin.

Although these findings are not novel [9,11,12,13,14,48,49,53], they confirm the presence of UVA- and heat-induced structural changes in the eumelanins and pheomelanins samples investigated by multiphoton FLIM imaging.

We characterized the fluorescence lifetime properties of the synthetic melanin samples in equivalent conditions to those used in our in vivo multiphoton clinical studies of human skin [26,28,30,34,37,43,54] and processed the 2PEF intensity decays with both the phasor analysis based on fast Fourier transform [26,50,51,52] and the bi-exponential fitting analysis [20,21,22,24,26]. We represented the data as phasor plots, allowing the regrouping of the melanin pixels with similar fluorescence lifetime properties and the visualization in one shot of the samples’ differences. Furthermore, using both methods, we quantified the different lifetime parameters and their relative fractions. For the phasor analysis, we introduced a new parameter called the relative concentration/fraction of UVA-modified melanins. Indeed, UVA-exposed melanins are a mixture of native melanins and UVA-modified (oxidized, photo-degraded, and crosslinked) melanins. Knowing one species allows the estimation of the fraction of the other species contributing to the global fluorescence signal, such as, for example, when determining the relative fractions of bound/free cellular metabolic coenzymes [55]. Using native melanins as reference samples, the $f_{UVA\;Mel}$ fraction was graphically measured as the distance of each experimental UVA-exposed melanin pixel in the phasor plot to the average location of the native melanins.

Our results globally indicate that native and UVA-exposed melanins are characterized by a mixed fluorescence species phasor fingerprint and exhibit a bi-exponential decay with a main relative contribution $a_{1}\left[\%\right]$ of the $\tau_{1}$ short fluorescence lifetime to the global fluorescence signal. Melanin solutions/suspensions contain polymers, and both the eumelanin and the pheomelanin samples were characterized by an approximately 2–3× shorter fluorescence lifetime $\tau_{1}$ compared to their monomers. At the supramolecular structure level, it is believed that DHI melanin (or more accurately DHI units in eumelanin) is stacked through π-π interaction between the DHI units and that the DHICA melanin is bundled through hydrogen bonding due to the presence of a carboxylate group in the DHICA units [56,57,58]. Apart from this point, little is known about the chemical nature of the supramolecular organization of melanins, especially pheomelanins. This supramolecular organization might shorten the fluorescence lifetime. This could be explained by a self-quenching phenomenon favored by the polymeric organization (melanin-to-melanin chromophores and Förster resonance energy transfer) and by the broad-band UV and visible absorption of melanin [59,60] (attenuation of the incident and the emitted light by the fluorophore itself). We think that this self-quenching could be increased by the aggregate-level organization of the melanin molecules in the polymer powders [61,62], which are probably organized as π-stacked sheets. The resonance energy transfer will not only take place between the individual chromophores of a sheet but also between the adjacent π-stacked sheets, thus possibly accounting for the absence of fluorescence or the presence of an instrumental response function type of signal detected in some powders.

**Multiphoton FLIM discriminates native eumelanins, pheomelanin, and mixed eu-/pheo-melanins**. For native melanins in solution, our data show that multiphoton FLIM parameters, mainly the short fluorescence lifetime, allows the discrimination between eumelanin, pheomelanin, and mixed eu-/pheo-melanins.

Both the DHI and the Dopa eumelanin suspensions have heterogenous phasor plots with a comet-like fingerprint pointing towards very short lifetime contributions, indicating the presence of both structures of the homogenous suspension of polymers and aggregates (small phase lifetimes). On the other hand, the DHICA eumelanin, pheomelanin, and mixed eu-/pheo-melanin solutions exhibited more homogenous phasor plots, with the highest *g* and the smallest *s* values, corresponding to very small phase lifetimes being measured for the DHICA eumelanin.

The DHI and DHICA eumelanins in solution, which are quite similar in terms of molecular structure, show very strong differences for all the FLIM phasor parameters and variable differences in the bi-exponential analysis parameters, probably due to the presence of the carboxylic group in the DHICA. Their lifetime properties are also different compared to those of Dopa eumelanin, which is often used in the literature as a standard for eumelanin.

Native pheomelanin (Dopa-Cys-1-1) in solution, a mixture of benzothiazine and benzothiazole at a 17.9 ratio, showed very strong differences in all the FLIM phasor parameters compared to the DHICA eumelanin, Dopa eumelanins, and mixed eu-/pheo-melanins, and compared to the DHI eumelanin, very strong differences were only evidenced for the phasor *g*, *s*, and $\tau_{m}$ modulation lifetime parameters. The differences using the FLIM bi-exponential fitting method varied from moderate to very strong depending on the parameters.

It was also important to characterize the mixed eu-/pheo-melanins (Eu/Pheo - Dopa-Cys-4-1 at a ratio of 75/25), as melanins in human skin are a mixture of DHI, DHICA eumelanins, and benzothiazine/benzothiazole pheomelanin [44,46]. We found that mixed eu-/pheo-melanins exhibited very strong differences in all the FLIM phasor parameters compared to the DHICA eumelanin, Dopa eumelanin, pheomelanin and strong to very strong differences compared to the DHI eumelanin. Using FLIM bi-exponential analysis, we found very strong differences for all the parameters compared to the DHICA eumelanin and native pheomelanin and parameter-dependent differences compared to the DHI eumelanin (only moderate differences in $\tau_{2}$ and $\tau_{Av\;Amp}$) and Dopa eumelanin (moderate for $\tau_{2}$ and very strong for the other parameters).

The multiphoton FLIM differences observed between the eumelanins, pheomelanins, and mixed eu-/pheo-melanins, as well as between the monomers and polymers and their aggregates, highlight the fluorescence lifetime dependency on the melanin structure and molecular organization.

These synthetic melanin multiphoton FLIM data are close to those obtained in situ for melanins within the human skin, hair, or eye [20,21,22,23,25,26]. For example, in human skin in vivo [26], melanin in the basal and supra-basal layers of the epidermis exhibited the shortest $\tau_{1}$ values with the highest $a_{1}\left[\%\right]$ relative contribution and were characterized by a phasor plot with a comet-like pattern pointing towards a very short lifetime component of around 0.1 ns (highest *g* and smallest *s* values). A similar phasor pattern was also evidenced in human choroidal melanocytes [25]. In these tissues, the multiphoton fluorescence lifetime properties of the melanins will depend on parameters such as the eu-/pheo-melanin ratio, their macromolecular organization, or the local environment of the melanosomes. This environment at the micrometer-scale level, within the two-photon excited sub-femtoliter volume, depends on the tissue organization: a mixture of melanins and keratins in human hair and within the human skin’s stratum corneum, compared to a mixture of melanins and NAD(P)H and FAD metabolic cofactors in the keratinocyte’s cytoplasm of the human skin and human hair bulb. Consequently, the multiphoton FLIM signal represents all the emitting fluorescent species found within the focal volume. To go further and extract the local eu-/pheo-melanin ratio from this signal, one could use the phasor approach for the multiphoton FLIM data analysis [23,25] in association with the reference native eumelanin and pheomelanin samples investigated in this work. These model melanin samples could potentially replace the current references, i.e., red and black human hairs corresponding, respectively, to mixed keratin/pheomelanin and keratin/pheomelanin/eumelanin species [45]. Moreover, using the reference mixed eu-/pheo-melanin solution, one could also extract the keratin to mixed melanins ratio or the free or bound NAD(P)H to melanin ratios in human skin cells and tissues.

**UVA exposure globally induces dose- and melanin type-dependent modulations in multiphoton FLIM lifetime parameters**. As demonstrated by the chemical analysis, UVA exposure induces structural changes in melanins, which were found in this study to impact their fluorescence lifetime properties. The 2PEF intensity decays showed differences between the native and UVA-exposed melanins, the smallest being observed for pheomelanin. These changes translated into a visible shift in the phasor fingerprint along with a modification of the phasor (*s*, *g*) pixel distribution from, for example, a heterogenous to a more homogenous fingerprint.

For the DHICA eumelanin solution, a progressive and very strong modulation of all the phasor parameters was observed with the UVA exposure. Although quite similar in terms of molecular structure to the DHICA eumelanin, the DHI eumelanin’s fluorescence lifetime response to UVA exposure was different, with a progressive and very strong increase in the $\tau_{\varphi }$ phase lifetime and a very strong increase in the $\tau_{m}$ modulation lifetime only after 7 days of UVA exposure. Strong to very strong modulations were also evidenced by the FLIM bi-exponential analysis. The Dopa eumelanin exhibited a different fluorescence lifetime response to the UVA dose, with a very strong increase in the $\tau_{\varphi }$ phase lifetime after 1 day and, to a lesser extent, at 7 days of UVA exposure and an equivalent very strong increase in the $\tau_{\varphi }$ phase lifetime with both UVA doses. With the FLIM bi-exponential analysis, very strong modulations were evidenced for the DHICA and strong to very strong modulations for the DHI and Dopa eumelanins. Globally, the UVA-exposed samples exhibited an increase in both fluorescence lifetimes, $\tau_{1}$ and $\tau_{2}$, with the highest modulation occurring in the short fluorescence lifetime, along with a decrease in its relative contribution $a_{1}\left[\%\right]$. The highest increase in fluorescence lifetime was observed for the DHI eumelanin.

The dose-dependent modulations of the FLIM parameters with the UVA exposure indicate a change in their structural organizations, probably due to a dose-dependent modulation of the ratio of the native to the UVA-modified (oxidized, photo-degraded, and crosslinked) compounds. Indeed, the HPLC chemical results show that the UVA exposure induced DHICA crosslinking (2.1× increase in PTeCA after 7 days) and oxidation (4.6× increase in free/total PTCA ratio after 1 day and 11.7× after 7 days). Changes in the eumelanin crosslinking PTeCA were also observed for the DHI eumelanin (increase of 1.2x; 1 day and 1.4x; and 7 days) and the Dopa eumelanin (a decrease after 1 day (1.3×) and an increase after 7 days (1.1×) of UVA exposure). The DHI or Dopa oxidation assessed by the free PTCA showed a 5.0× increase in the DHI eumelanin and 3.0× in the Dopa eumelanin after 7 days, although the absolute values were small.

Using the phasor parameter of the $f_{UVA\;Mel}$ relative fraction of the UVA-modified melanin species, implemented here for the first time, we found a progressive and very strong increase in this fraction with the UVA dose for the DHICA eumelanin, which exhibited the highest UVA-induced modulation, followed by the mixed eu-/pheo-melanins, DHI eumelanin, Dopa eumelanin, and, to a lesser extent, by pheomelanin. In Dopa eumelanin and pheomelanin, the highest modulation was observed after 1 day of UVA exposure.

Pheomelanin (Pheo - Dopa-Cys-1-1), a mixture of benzothiazine and benzothiazole, showed the weakest modulations in both the phasor and the bi-exponential analysis parameters, with moderate, strong, and very strong differences depending on the parameters. A very strong increase in the $\tau_{\varphi }$ phase and $\tau_{m}$ modulation lifetimes and $f_{UVA\;Mel}$ relative fraction was measured after 1 day of UVA exposure, whereas at 7 days of exposure this fraction was found to moderately decrease. Small changes in the bi-exponential lifetime parameters were also detected (no change in $\tau_{1}$ after 1 day, but a strong decrease after 7 days; $a_{1}\left[\%\right]$ strongly decreased at 1 day; and a moderate increase and decrease, respectively, in $\tau_{2}$ with the UVA dose). These changes translated into a strong increase at 1 day in both the combined lifetime parameters, $\tau_{Av\;Int}$ and $\tau_{Av\;Amp}$.

The UVA light induced pheomelanin oxidation, i.e., the partial conversion of benzothiazine into benzothiazole, as indicated by the chemical analysis HPLC pheomelanin oxidation ratio (TTCA benzothiazole/4-AHP benzothiazine). The slight UVA modulations observed for pheomelanin suggest that the two types of pheomelanins probably have similar fluorescence characteristics. Native benzothiazine and benzothiazole pheomelanin polymers cannot be synthetized separately to verify this hypothesis. Nevertheless, the multiphoton FLIM data acquired on the monomers of benzothiazine and benzothiazole indicate that their fluorescence lifetimes are similar. Thus, we can extrapolate that the two types of pheomelanin polymers have similar multiphoton FLIM properties in the investigated solution and that one cannot use this method to assess their UVA-induced modifications.

UVA-induced modulations in the fluorescence lifetime parameters were also detected in the mixed eu-/pheo-melanins (Dopa-Cys-4-1 at a ratio of 75 (1.36× more DHICA (PTCA) than DHI (PDCA))/25 (10.21× more benzothiazine (4-AHP) than benzothiazole (TTCA) pheomelanins). All the phasor and bi-exponential analysis parameters were very strongly modulated after 7 days of UVA exposure. We evidenced a very strong increase in the $\tau_{\varphi }$ phase and $\tau_{m}$ modulation lifetimes and in the $f_{UVA\;Mel}$ relative fraction of the UVA-modified species. The short $\tau_{1}$ and the long $\tau_{2}$ fluorescence lifetimes increased very strongly, while the relative contribution $a_{1}\left[\%\right]$ very strongly decreased, resulting in a very strong increase in both the combined lifetime parameters, $\tau_{Av\;Int}$ and $\tau_{Av\;Amp}$.

The slight UVA changes measured for pheomelanin suggest that the modifications in the mixed eu-/pheo-melanins are probably driven by eumelanin changes. Conversely, the chemical analysis HPLC results indicate no change or very slight modifications in the DHI and DHICA eumelanin crosslinking and oxidation and an increase in oxidized pheomelanin (1.47× more benzothiazole). Moreover, upon UVA exposure, the decrease in native benzothiazine content was 2.44× higher than its conversion to benzothiazole. This could be explained by the benzothiazine oxidation to benzothiazole pheomelanin, affording TDCA, and also by TTCA photo-oxidation to unidentified products. For this type of mixed melanins, the multiphoton FLIM and HPLC results seem to provide different information. However, one should keep in mind firstly that the HPLC method requires high amounts of melanin for analysis (e.g., for human skin, two biopsies of 0.8 cm^2^ [44]) compared to the sub-femtoliter volume (~0.32 µm^3^; smaller than the melanosome size) of the multiphoton FLIM method. Secondly, the synthetic mixed eu-/pheo-melanin solution cannot be prepared with enough quantities of DHICA units for it to be possible to detect the modulations related to eumelanin’s oxidation and crosslinking by the HPLC method. Therefore, multiphoton FLIM analysis seems to be more sensitive in detecting the global fluorescence lifetime changes induced by UVA exposure in this type of mixed melanins, whether these changes are due to eumelanin or pheomelanin.

Regardless of the type of modifications, overall the multiphoton FLIM imaging could highlight the UVA-induced changes in this type of mixed eu-/pheo-melanins. This result is promising for the in vivo assessment of UVA-induced modifications in human skin, as this type of tissue contains mixed eu-/pheo-melanins at a similar ratio [44] to that in our synthetic solution.

The increase in the fluorescence lifetime could be due to melanin fragmentation and degradation upon UVA exposure [9,12,13,14]. Such a photo-degradation may increase the distances between the individual melanin chromophores, which will be less involved in the melanin-to-melanin energy transfer, leading to a reduction in the fluorescence quenching, probably by disrupting the supramolecular organization, that is, the interaction between the π-stacked sheets of DHI units in the context of the eumelanin. This is only an assumption, but one can imagine that the supramolecular organization of the UVA-exposed samples compared to that of the native samples is impacted by the presence of photo-degraded and crosslinked melanin polymers. Whatever the mechanism behind the UVA-induced structural changes, they translate into modulations of the fluorescence lifetime properties of the melanins.

Altogether, these results demonstrate the ability of multiphoton FLIM imaging, coupled with phasor and bi-exponential analysis, to evidence the global structural changes appearing at the global level in eumelanins and mixed eu-/pheo-melanins upon UVA exposure. According to the HPLC results, native DHICA eumelanin is partially converted by UVA light to crosslinked and oxidized eumelanins, and crosslinking changes were also evidenced for the DHI and Dopa eumelanins. The lack of fluorescence signals from the native and crosslinked melanin powders prevented us from assigning the FLIM modulation to the crosslinked or oxidized melanins.

## 4. Materials and Methods

### 4.1. General Information on Melanin Types, Their UVA-Induced Modifications, and Chemical Analysis Using the HPLC Method

Melanins are a mixture of eumelanins (insoluble black to brown pigment) and pheomelanins (alkaline-soluble, yellow to reddish-brown pheomelanin) [56,63,64,65,66], a collection of molecular species with different chemical structures and properties encompassing diverse degrees of chemical heterogeneity, from their monomer unit structure to the different modes of inter-unit coupling, supra-molecular organization, and aggregation [56,57,58,63,64,67].

Both melanin types are derived from dopaquinone, which is produced from tyrosine by the action of tyrosinase [68], but the availability of cysteine determines the type of pigment, i.e., pheomelanin or eumelanin, produced in the melanosomes [69,70] (Figure 7a).

Cysteine’s reaction with dopaquinone produces cysteinyldopas (5-*S*-cysteinyldopa (5SCD) and 2-*S*-cysteinyldopa (2SCD) in a ratio of 5.3:1), which are further oxidized by dopaquinone to produce benzothiazine intermediates that will gradually polymerize to form the pheomelanin pigment. In the late production stage, the benzothiazine pheomelanin is gradually converted to benzothiazole pheomelanin [71], and together, they form a mixed benzothiazine/benzothiazole pheomelanin polymer.

Upon cysteine’s depletion within the melanosomes, dopaquinone spontaneously reacts to give, via dopachrome, DHI and DHICA. The production of DHICA eumelanin is accelerated by tyrosinase-related protein 2 (dopachrome tautomerase) or copper ions [72]. These 5,6-dihydroxyindoles are then further oxidized to produce the mixed DHI/DHICA eumelanin polymer.

Human skin melanin polymers are a mixture of eu-/pheo-melanins (DHI and DHICA eumelanins and benzothiazine and benzothiazole pheomelanins). The eu-/pheo-melanins ratio was quantified to be approximately ~74% eumelanin to 26% pheomelanin (mostly of benzothiazole type) in human breast skin samples of variable pigmentation [44,46].

UVA or blue light exposure has been shown to induce the partial photo-oxidative degradation/modification of melanins [12,13,14,15,16] (Figure 7b). Natural and synthetic eumelanins (DHI and DHICA moieties) undergo oxidative cleavage of the indolequinone moiety to give free PTCA, a product of DHICA peroxidation, and photo-degraded eumelanin with the putative diaryl ketone structure [9,12], a highly fluorescent compound [12] whose formation was evidenced upon the photo-oxidation of human retinal pigment epithelium melanosomes [73,74]. The reaction of the DHI (and/or DHICA) moiety with the indolequinone moiety gives rise to eumelanin crosslinking [13] (Figure 7b), which can also be obtained by heating [48,49] (Figure 7c). Natural and synthetic pheomelanins also undergo degradation/modification upon UVA exposure (Figure 7d), i.e., oxidative conversion of the benzothiazine moiety to the benzothiazole moiety [11,12,53].

Native or photo-induced modifications of eumelanins and pheomelanins can be investigated by the HPLC chemical analysis of melanin degradation products, obtained after AHPO or reductive hydrolysis with HI [10,11].

The AHPO of eumelanin gives rise to specific degradation products of the DHICA (PTCA) and DHI (PDCA) moieties [11,75] (Figure 7c).

Reductive hydrolysis with hydroiodic acid of pheomelanin pigment gives 4-AHP and 3-AHP degradation products [10,75,76] (Figure 7d). 4-AHP and 3-AHP arise, respectively, from the cysteinyldopas, 5SCD- and 2SCD-derived benzothiazine moieties. The benzothiazole moiety can be analyzed after AHPO via the TTCA and TDCA degradation products [11] (Figure 7d). Upon UVA exposure, benzothiazine pheomelanin is converted into benzothiazole pheomelanin and the TTCA/4-AHP ratio is used as a marker for pheomelanin photo-degradation [11,12].

Upon exposure to UVA, eumelanin is photo-degraded and crosslinked. The product of eumelanin crosslinking is analyzed after AHPO by estimation of the PTeCA that appears to be derived from the crosslinking of the C2 and C3 positions of the DHI moiety in eumelanin [13] (Figure 7c).

The ratio of PTeCA/PTCA is indicative of eumelanin crosslinking, while the ratio of free/total PTCA is indicative of eumelanin photo-degradation (Figure 7b).

### 4.2. Synthetic Melanin Samples

Table 3 describes the different types of investigated melanin samples, in solution, suspension, or powder. We studied (i) native melanins; (ii) melanins exposed to UVA light at a radiance of 3.5 mW/cm^2^ for 1 and 7 days to induce different levels of melanin photo-degradation, C-C crosslinking, and some decarboxylation; and (iii) melanins exposed over 8 days at a temperature of 100 °C to induce melanin crosslinking and extensive decarboxylation [48]. Synthetic melanin samples were prepared according to the previously published protocols: monomers [66,71], polymers in solution/suspension, and polymers in powders [66]. The UVA exposure of the melanin solution/suspension was performed for 24 h and 7 days at a radiance of 3.5 mW/cm^2^ (Oriel 3000 W solar simulator), which is similar to the solar radiance in Greece in June [47]). The long exposure period was chosen in order to maximize the UVA-induced modifications in melanins. For the heating experiments, the melanin preparations were ground to a fine powder using an agar mortar and pestle and were heated at 100 °C under argon in a capped glass tube for 8 days [74].

#### 4.2.1. DHICA Eumelanin (Eu - DHICA)

Eu - DHICA melanin corresponds to native DHICA eumelanin and was prepared by enzymatic oxidation of DHICA. After UVA exposure, eumelanin is partially oxidized: the final solution contains both native and oxidized (photo-degraded and C-C crosslinked) eumelanins. Photo-degraded DHICA eumelanin cannot be isolated, but crosslinked DHICA eumelanin can be obtained by heating the Eu-DHICA powders at 100 °C for 8 days. DHICA melanin was prepared by oxidizing 1 mM DHICA in sodium phosphate buffer, pH 6.8, by mushroom tyrosinase (100 U/mL) at 37 °C for 4 h.

#### 4.2.2. DHI (Eu - DHI) and Dopa (Eu - Dopa) Eumelanins

Eu - DHI and Eu - Dopa correspond to native DHI and Dopa eumelanins, respectively. DHI eumelanin was prepared by the oxidation of DHI by mushroom tyrosinase, whereas Dopa eumelanin was prepared by the oxidation of L-Dopa by mushroom tyrosinase. Dopa eumelanin is considered to be the melanin gold standard although it is not completely identical to typical natural eumelanin as synthetic Dopa eumelanin consists mostly of DHI (DHI:DHICA ratio of ca. 9:1). These samples were also exposed to UVA light and heating in order to induce different types of oxidations. DHI and Dopa melanins were prepared by oxidizing 1 mM DHI or 1 mM Dopa in sodium phosphate buffer, pH 6.8, by mushroom tyrosinase (100 U/mL) at 37 °C for 4 h.

#### 4.2.3. Mixed DHI and DHICA (Eu - DHI + DHICA 1-1) Eumelanins

Eu - DHI + DHICA 1-1 corresponds to native mixed DHI and DHICA eumelanins at a ratio of 50/50. This type of mixed eumelanins was prepared by the oxidation of a 1:1 molar ratio of DHI and DHICA to provide another type of reference eumelanins (natural eumelanin consists of various ratios of DHI and DHICA eumelanins).

#### 4.2.4. Pheomelanin (Pheo - Dopa-Cys-1-1)

Pheo - Dopa-Cys-1-1 corresponds to native pheomelanin (mainly benzothiazine pheomelanin) and is a mixture of 5-*S*-cysteinyldopa and 2-*S*-cysteinyldopa in a ratio of ca. 5:1. This pheomelanin was prepared from a 1:1 molar ratio of L-Dopa: L-cysteine [66]. The UVA-oxidized samples contain both native (benzothiazine) and oxidized (benzothiazole) pheomelanins.

#### 4.2.5. Benzothiazine and Benzothiazole Pheomelanin Monomers

The native pheo-benzothiazine monomer (7-(2-amino-2-carboxyethyl)-5-hydroxy-3,4-dihydro-2*H*-1,4-benzothiazine-3-carboxylic acid - DHBTCA) and pheo–benzothiazole monomer (6-(2-amino-2-carboxyethyl)-4-hydroxybenzothiazole - BZ-AA) were prepared according to the methods described in [71] to serve as model compounds for the multiphoton FLIM experiments. Pheo-benzothiazine is colorless when prepared, but the compound is rapidly oxidized to give a blue trichochrome pigment.

#### 4.2.6. Mixed 75/25 Ratio eu-/pheo-melanins (eu/pheo - Dopa-Cys-4-1)

Eu/Pheo - Dopa-Cys-4-1 corresponds to native mixed eu-/pheo-melanins at a ratio of ca. 75/25 which are found in the human epidermis [44]. This type of mixed eu-/pheo-melanins was prepared in a solution of L-dopa (1.97 mg) and L-cysteine (0.30 mg) in a 10 mL buffer and was not isolated. Therefore, the concentration was assumed to be 0.227 mg/mL. In this suspension, eumelanin is composed of 10% DHICA and 90% DHI [72]. The UVA-oxidized mixed melanins are a mixture of both native and oxidized (photo-degraded and crosslinked) DHI and DHICA eumelanins and benzothiazine and benzothiazole pheomelanins.

#### 4.2.7. Mixed 50/50 Ratio eu-/pheo-melanins (eu/pheo - Dopa-Cys-2-1)

Eu/Pheo - Dopa-Cys-2-1 corresponds to the native mixed eu-/pheo-melanins at a ratio of ca. 50/50. This type of mixed eu-/pheo-melanins was prepared by the oxidation of a 2:1 molar ratio of L-dopa and L-cysteine and can be considered as a 1:1 molar ratio of eumelanin and pheomelanin.

### 4.3. Chemical Analysis of Melanin Degradation Products—HPLC Method

The eumelanin and pheomelanin contents were indirectly estimated by the HPLC analysis of the specific degradation products obtained after alkaline hydrogen peroxide oxidation (AHPO), K_2_CO_3_ extraction, and hydroiodic acid (HI) hydrolysis. For this analysis, a quantity of 0.1 mL of each melanin solution/suspension was used, and the results were similar to those reported in [12]. For the powder samples, the melanin was ground, and a suspension of a 2 mg/mL concentration in water was prepared. Aliquots of 0.1 mL were subjected to AHPO and HI hydrolysis.

### 4.4. Multiphoton FLIM Imaging

Multiphoton 2PEF-FLIM imaging was performed using a commercial confocal/multiphoton FLIM microscope (Leica TCS SP8 MP FLIM, Leica Microsystems CMS GmbH, Mannheim, Germany) equipped with a TCSPC (time-correlated single-photon counting) FLIM module (Picoquant, Berlin, Germany).

Prior to imaging, the samples were defrosted at room temperature, followed by a homogenization step (vortex) for melanin solutions or suspensions. The samples (a few µg for powders and ~20 µL for solutions) were placed afterwards between a glass slide (VWR) and a glass coverslip (Marienfield 24 × 50 mm^2^) using an adhesive silicon isolator (JTR8R-0.5 8 × 9 × 0.5 mm^3^). The glass slides were cleaned with alcodox 2% in an ultrasound water bath, rinsed with water and ethanol and dried with pressurized air.

For each sample, we investigated 3 to 4 different regions. In each region, a multiphoton FLIM image was acquired upon two-photon excitation at 760 nm using an IR pulsed femtosecond laser (InSight X3, Spectra-Physics^®^, Santa Clara, CA, USA; laser frequency: 80,08 MHz, pulse width < 120 fs) and a water immersion 25x/0.95NA Leica HC FLUOTAR L objective. The 2PEF-FLIM signal in the 409–680 nm range was epi-detected using a hybrid detector HyD-RLD1 in the photon counting mode. Each image of 204.08 × 204.08 µm^2^ (512 × 512 pixels × 783 time channels, integration time 16 ps/time channel, dx = 399.38 nm, 1.2 µs/pixel, 20 frame accumulations, ~26 s/image) was acquired using an excitation power of 21 mW for the melanin solutions and 2 mW for the powders.

### 4.5. Bi-Exponential Fitting Analysis

The FLIM parameters were computed by fitting the fluorescence intensity decay with a bi-exponential function [20,21,22,24], $I_{2PEF}\left(t\right)=a_{1}e^{-\frac{t}{\tau_{1}}}+a_{2}e^{-\frac{t}{\tau_{2}}}$, to extract the short and long fluorescence lifetimes $\tau_{1}$ and $\tau_{2}$ and their relative amplitudes $a_{1}\left[\%\right]=\frac{a_{1}}{a_{1}+a_{2}}$ and $a_{2}\left[\%\right]=\frac{a_{2}}{a_{1}+a_{2}}$ and to compute the amplitude-weighted $\tau_{AvAmp}=\frac{a_{1}\tau_{1}+a_{2}\tau_{2}}{a_{1}+a_{2}}$ or intensity-weighted $\tau_{AvInt}=\frac{a_{1}{\tau_{1}}^{2}+a_{2}{\tau_{2}}^{2}}{a_{1}\tau_{1}+a_{2}\tau_{2}}$ average lifetimes. Simple fluorescent molecules have single exponential decays (e.g., simulated decays; see Figure 8a), whereas endogenous fluorophores such as melanin exhibit a bi-exponential decay similar to the simulated A&B mixed species decay (Figure 8a). FLIM bi-exponential analysis was performed using SymPhoTime 64 (v2.1.3813, Picoquant, Berlin, Germany) software. Images were spatially (20 × 20 pixels) and temporally (2 time channels) binned to increase the signal intensity per time channel, and the FLIM parameters were calculated using a 2-exponential reconvolution model and a calculated instrumental response function (IRF). The calculated parametric images were exported in tif format and further quantified with Fiji/ImageJ (v1.54b, W. Rasband, National Institutes of Health, Bethesda, MD, USA). For each parameter image, a jet color lookup table was employed, and 4 size-equal regions of interest (ROIs) were defined. The mean parameter values of each ROI, estimated for all the pixels with a 2PEF signal intensity above the noise level, and a chi^2^ error fitting value less than 2 were considered for data statistical analysis.

### 4.6. Phasor Analysis

By applying a Fourier transform to every pixel, the phasor method [50,51,52] transforms the 2PEF intensity decay into a phasor with coordinates *g* and *s* within the phasor plot (Figure 8b). The real *g* and complex *s* components of the cosine and sine transforms, respectively, are defined by $g_{i,j}\left(\omega \right)=\frac{\int_{0}^{\infty }I_{i,j}\left(t\right)\cos\left(\omega t\right)dt}{\int_{0}^{\infty }I_{i,j}\left(t\right)dt}$; $s_{i,j}\left(\omega \right)=\frac{\int_{0}^{\infty }I_{i,j}\left(t\right)\sin\left(\omega t\right)dt}{\int_{0}^{\infty }I_{i,j}\left(t\right)dt}$ where *i*, *j* are the pixel coordinates in the FLIM image, ω = 2π*f* is the angular frequency, and *f* is the laser repetition rate (80,08 MHz in our data). Parameters *g* and *s* can be expressed in terms of *m* modulation and *φ* phase angle (Figure 8b), which are used to estimate the apparent phase $\tau_{\varphi }=\frac{1}{\omega }\tan\left(\varphi \right)$ and modulation $\tau_{m}=\frac{1}{\omega }\sqrt{\left(\frac{1}{m^{2}}-1\right)}$ lifetimes.

For single exponential decays, these two lifetimes are equal, and the phasor coordinates *g* and *s* lie on the universal semicircle with radius 0.5 going from point (1, 0) to point (0, 0), corresponding, respectively, to τ=0 and τ=∞. Conversely, the phasor coordinates of the mixed A&B species lie inside the universal semicircle along a line connecting the two distinct lifetime phasors (A and B). The phasor plot of an *n*-component mixture will reside within a polygon with *n*-vertices located in the position of the phasor of each contributing species. Its coordinates are given by $G\left(\omega \right)=\underset{n}{\sum }f_{n}g_{n}\left(\omega \right)$ and $S\left(\omega \right)=\underset{n}{\sum }f_{n}s_{n}\left(\omega \right)$, where the relative contributions are normalized, $\underset{n}{\sum }f_{n}=1$.

The two-photon excited fluorescence signal of melanin decays as a sum of two exponentials, and therefore, its phasor plot lies within the universal semicircle and reflects the mixed molecular species contributions. Image pixels with similar 2PEF intensity decays and similar *s* and *g* phasors will be clustered in the phasor plot.

Phasor analysis was performed with Flute (Fluorescence Lifetime Ultimate Explorer) (v.1.0.0, Laboratory for Optics and Biosciences, Palaiseau, France) [77]. Prior to phasor analysis, FLIM images in ptu format were converted to tif format using the Fiji/ImageJ ptu reader v.0.0.9 plugin (Cell Biology group, Utrecht University, Utrecht, The Netherlands) [78] and spatially (10 × 10 pixels) binned to increase the signal-to-noise ratio. Phasor data calibration was performed using the 2PEF FLIM data of a fluorescein sodium salt—CAPS solution (reference 67884, Sigma-Aldrich/Merck KGaA, Darmstadt, Germany), acquired in the same condition as the melanin samples and at 1.8 mW laser mean power.

We first calculated the *g*, *s*, phase, and modulation lifetimes parameters. Secondly, we estimated the average *g* and *s* values of all the images acquired within a native melanin solution. These values were afterwards used to calculate the $f_{UVA\;Mel}$ relative fraction of the UVA-modified eumelanin species (see Figure 6e). In our study, the melanin samples exposed to UVA are a mixture of two species: “native melanins” and “melanins structurally modified by the UVA exposure”. The $f_{UVA\;Mel}$ parameter was calculated as the phasor plot distance of every UVA-exposed experimental data point to the reference native melanin average *g*, *s* position (Figure 4e), in a similar manner to the calculation of the *f*_B_ fraction in Figure 8.

All parameters were calculated without thresholds or median filters and were exported as color coded tif files and raw tiff files that were further quantified with Fiji/ImageJ (v1.54b, W. Rasband, National Institutes of Health, Bethesda, MD, USA). The mean parameter values, estimated within the same ROIs as for the FLIM bi-exponential analysis parameters, were considered for data statistical analysis.

### 4.7. Statistical Analysis

Descriptive statistical analysis was performed with OriginPro 2022 (OriginLab, Northampton, MA, USA). The data were represented as bar plots with mean, median, and confidence intervals of the mean and dots for the raw data points. Comparisons between the native melanin types and the control and the UVA exposure conditions were performed using contrasts with both Student t-tests and Cohen-D effect size (ES, the ratio of the within-group difference to the square root of the variance of the sum of the random parameters in the model), using a homemade R-based software package (My Explorer v.0.6.2.8, L’Oréal, Aulnay-sous-Bois, France). The effect size characterizes the strength of the modifications observed between the groups and depends only on the underlying population parameters and not on the sample size as the *p*-value.

## 5. Conclusions

To better understand the impact of sunlight exposure on human skin, the chemical characterization of native melanins and their structural photo-modifications is of central interest. As the methods used today are invasive, we investigated the possibility of using multiphoton FLIM imaging along with both phasor and bi-exponential fitting analyses to differentiate native eumelanins, pheomelanins, and mixed eu-/pheo-melanins, as well as to detect their UVA-induced structural modifications.

HPLC chemical analysis was used as a reference method enabling the characterization of native melanins and their modifications induced by heating (crosslinked melanins) and UVA exposure (photo-degraded, oxidized, and crosslinked melanins). The UVA-exposed samples contained different levels of mixtures of both the native melanin species and the UVA-modified species, depending on the UVA dose (1 day or 7 days of exposure). Long exposure periods of 1 day and 7 days at 3.5 mW/cm^2^ UVA radiance were chosen in this study in order to maximize the UVA-induced modifications in the melanins.

We demonstrated that the multiphoton FLIM phasor and bi-exponential analysis methods allow the discrimination between the native DHI, DHICA, Dopa eumelanins, pheomelanin, and the mixed 75/25 ratio of eu-/pheo-melanin polymers in solution/suspension. We provide for the first time, to our knowledge, the multiphoton FLIM characteristics of “pure” pheomelanin, DHICA, and DHI eumelanins. These multiphoton FLIM data of native and mixed melanins could serve as reference data for the phasor analysis of melanin-containing cells, hair, and skin samples, instead of the commonly mixed keratin/pheomelanin and keratin/pheomelanin/eumelanin species in, respectively, red and black hair samples [23,25]. Moreover, using the reference mixed eu-/pheo-melanin solution, one could also extract the keratin to mixed melanin ratio or the free or bound NAD(P)H to melanin ratios in human skin cells and tissues.

We also provided evidence, for the first time, that the fluorescence lifetime properties of eumelanins, pheomelanins, and mixed eu-/pheo-melanins are modulated in a melanin-dependent and dose-dependent manner by UVA exposure, with the strongest modifications being observed for DHICA eumelanin and the weakest for pheomelanin. The UVA-induced crosslinking, photo-degradation, and oxidative changes in the chemical structure of melanins were globally evidenced by multiphoton FLIM via an increase in the fluorescence lifetimes, along with a decrease in their relative contributions, which was probably due to a decreased fluorescence quenching favored by the UVA-disrupted supramolecular organization of the melanin polymers. Moreover, we introduced a new phasor parameter of the relative fraction of a UVA-modified species, calculated using the native melanins as reference samples, and provided evidence for its sensitivity in assessing the UVA-induced structural modifications in the different melanin types.

Multiphoton FLIM imaging and bi-exponential/phasor data analysis, in association with appropriate eu-/pheo-melanin reference samples, hold promising perspectives for in vivo human skin mixed melanin characterization and the assessment of their fluorescence lifetime changes, such as with pigmentation genotypes. Another very exciting application could be the in vivo study of the UVA effects on human skin mixed melanins. What are the changes occurring in the epidermal melanin (3D density, z-epidermal distribution, and fluorescence lifetime parameters) in human volunteers exposed to UVA radiation? What is the impact of UVA light on in vivo mixed melanin fluorescence lifetime properties? Are they in agreement with the in tubo results? Is multiphoton FLIM imaging able to evidence melanin photo-degradation, oxidation, and crosslinking changes in vivo? We already demonstrated that multiphoton imaging is able to measure in vivo the melanin density modulations appearing with the seasonality/sunlight exposure of human forearm skin [30], but are there any modulations of melanin fluorescence lifetime parameters appearing with sunlight exposure? There are so many questions that need to be addressed, and there is an exciting journey to be taken in the world of melanins, which have not yet revealed all their secrets.

## Figures and Tables

**Figure 1 ijms-24-04517-f001:**
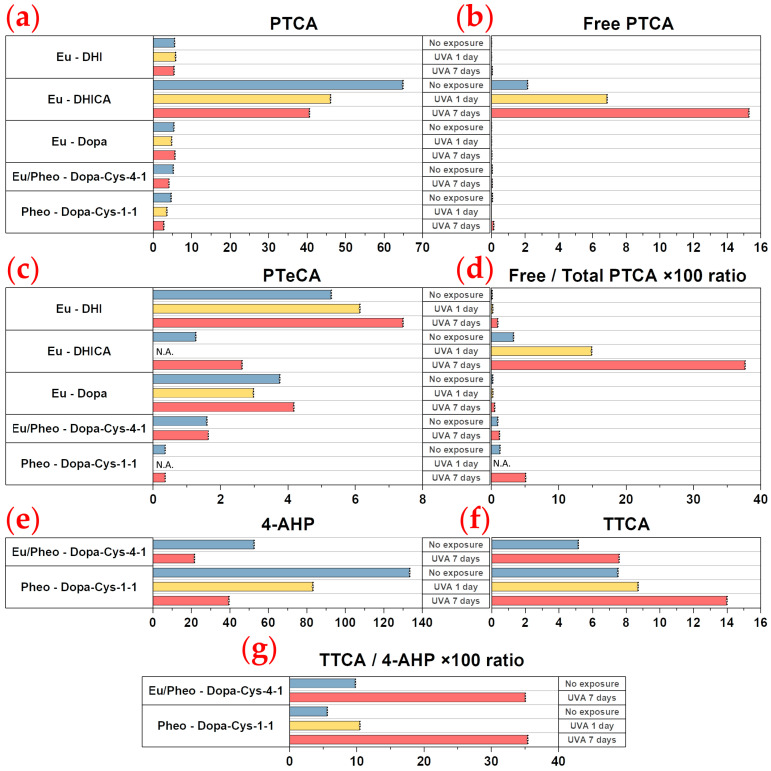
Chemical degradation—HPLC quantification results showing the effects of 1 and 7 days of UVA exposure at 3.5 mW/cm^2^ in eumelanins (Eu-DHICA, Eu-DHI, and Eu-Dopa), pheomelanin (Pheo - Dopa-Cys-1-1), and mixed eu-/pheo-melanin (Eu/Pheo - Dopa-Cys-4-1 at a ratio of 75/25) samples. (**a**) PTCA characterizes native DHICA eumelanin; (**b**) free PTCA the DHICA eumelanin peroxidation; (**c**) PTeCA the eumelanin crosslinking; and (**d**) free/total PTCA ratio the DHICA eumelanin oxidation. The (**e**) 4-AHP parameter characterizes benzothiazine pheomelanin and (**f**) TTCA the benzothiazole pheomelanin. (**g**) TTCA/4-AHP ratio reflects the pheomelanin oxidation, i.e., the conversion of benzothiazine to benzothiazole. All quantities are in µg/mg, except for the ratio parameters. N.A. indicates unavailable data.

**Figure 2 ijms-24-04517-f002:**
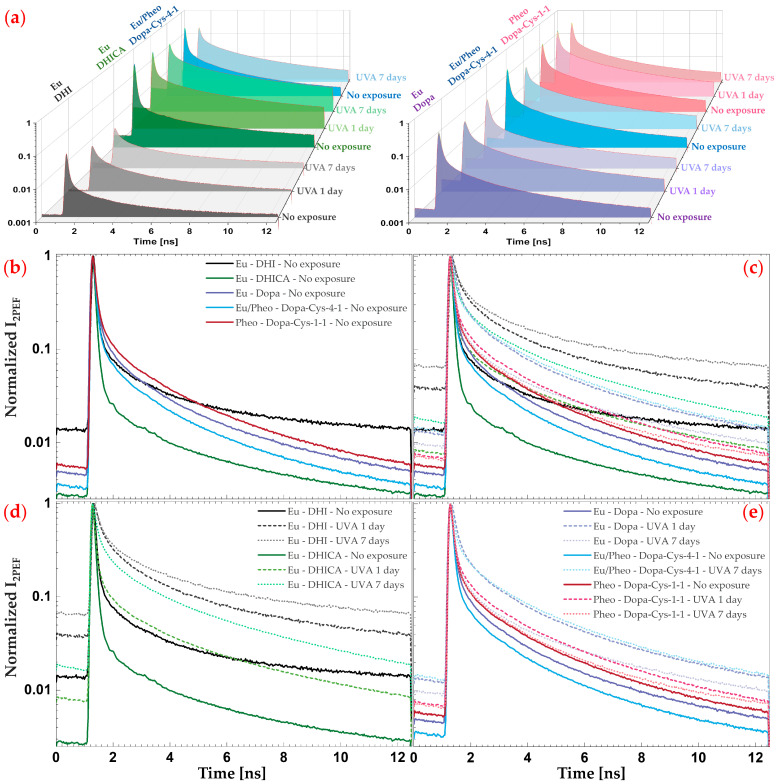
Multiphoton FLIM—example of normalized 2PEF intensity decays of native melanins and melanins exposed to 1 and 7 days of 3.5 mW/cm^2^ UVA. Each decay is an overall decay over one 512 × 512 pixel image, spatially (20 × 20 pixels) and temporally (2 time channels) binned. In (**a**) the fluorescence signals were normalized compared to native DHICA eumelanin to visualize signal intensity differences between conditions, whereas in (**b**–**e**) each decay was normalized to its maximum intensity to visualize the differences in the shape of the decays. The 2PEF intensity decays are given for (**b**) native melanins; (**c**) native and UVA-exposed melanins; (**d**) native and UVA-exposed eumelanins; and (**e**) native and UVA-exposed Dopa melanin, pheomelanin and mixed eu-/pheo-melanins. The color legend in (**c**) is the same as in (**b**,**d**,**e**).

**Figure 3 ijms-24-04517-f003:**
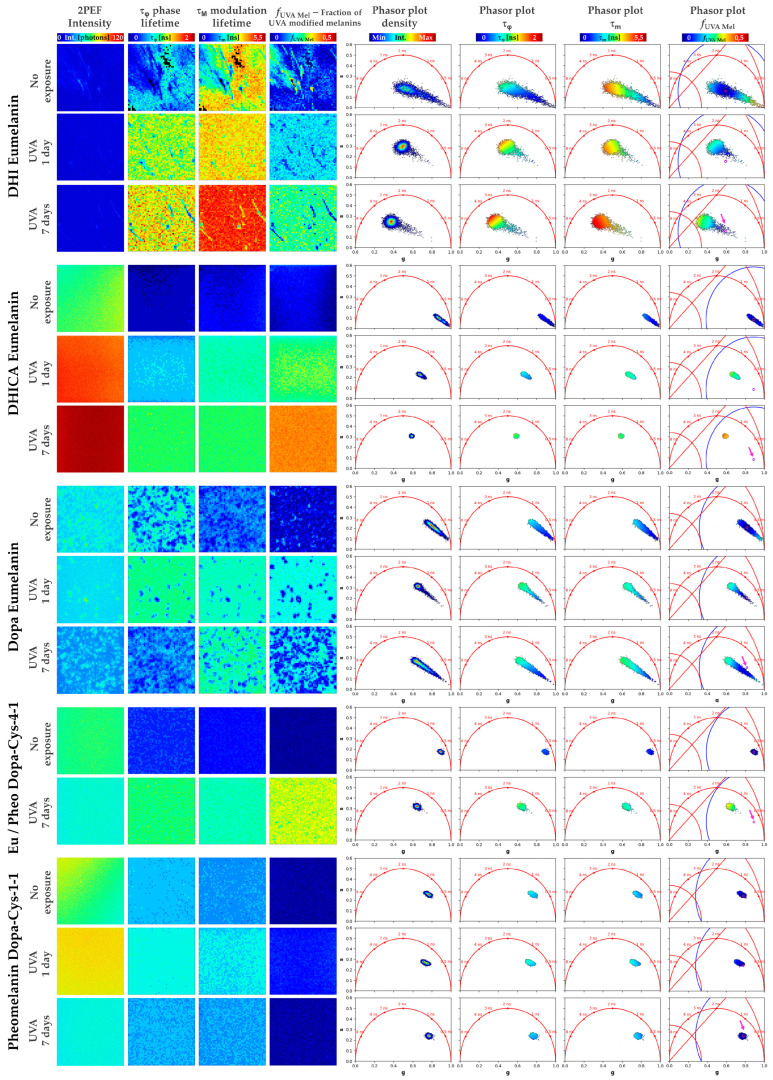
Multiphoton FLIM phasor analysis of native and UVA-exposed eumelanins, pheomelanin, and mixed eu-/pheo-melanins. (**left**) For each condition, an example of a raw 2PEF intensity (256 × 256 pixels) image is shown on the left, followed by the calculated phasor images (10 × 10 pixels spatial binning) of phase lifetime, modulation lifetime, and fraction of UVA-modified melanin parameters. (**right**) Phasor plots (scatters of phasor *s* versus *g* parameters) of all the images acquired per condition, allowing the visualization of the native melanins’ FLIM fingerprints and their change with UVA exposure. Every fluorescence intensity decay (e.g., Figure 2) of every pixel of an image is represented by a phasor pixel in the phasor plot with *s* and *g* coordinates. Pixels with similar 2PEF intensity decays will have close *s* and *g* coordinates and thus will be regrouped. Each phasor plot is color coded by a different phasor parameter and is correlated with its corresponding phasor image: every pixel in the phasor plot can be traced back to the pixel with the same property in the image. The density phasor plots highlight the pixel intensities (the fluorescence intensity is adjusted to the minimum and maximum of each image), whereas the $\tau_{\varphi }$, $\tau_{m}$, and $f_{UVA\;Mel}$ phasor plots allow identification of the pixels based on the values of these parameters. In the $f_{UVA\;Mel}$ phasor plots, the pink circle (see arrows) indicates the average *s* and *g* coordinates of native melanins. For every pixel, the fraction of UVA-modified melanins is calculated by measuring its distance to this reference native melanin position. The arcs and lines shown on the rightmost phasor plots indicate the min and max values of the color scales used for each parameter: the blue arc corresponds to the 0.5 fraction value limit, the red arcs to the modulation lifetime from 0 to 5.5 ns, and the red line to the 2 ns phase lifetime value limit.

**Figure 4 ijms-24-04517-f004:**
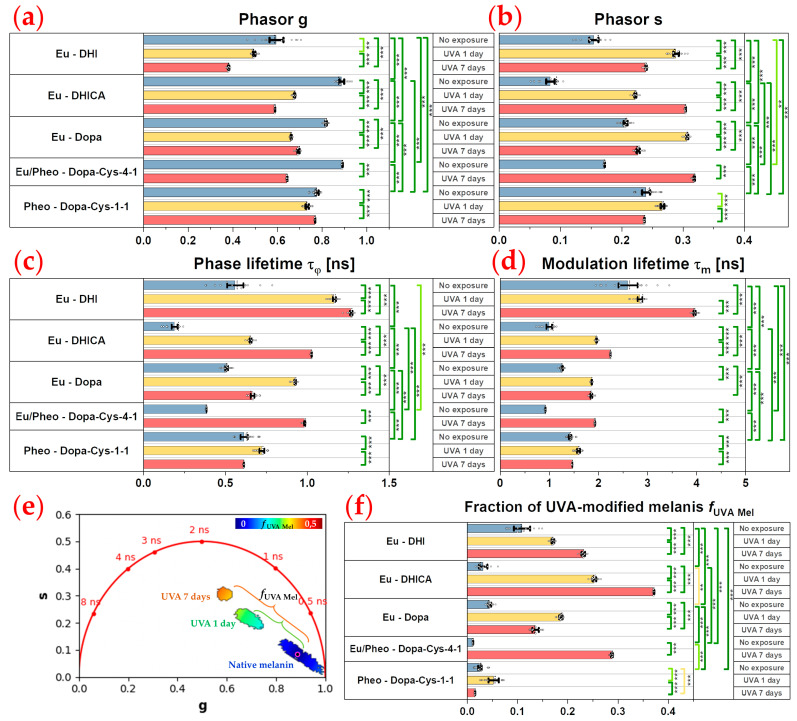
Multiphoton FLIM phasor analysis quantification results of native and UVA-exposed eumelanins, pheomelanin, and mixed eu-/pheo-melanins. For each type of melanin sample and exposure condition and every image pixel, we quantified the phasor parameters (**a**) *g* and (**b**) *s* (respectively, the real and complex components of the Fourier transform of the 2PEF intensity decay), (**c**) the phase $\tau_{\varphi }$, and (**d**) the modulation $\tau_{m}$ lifetimes. The diagram in (**e**) for DHICA eumelanin illustrates the relative fraction of UVA-modified melanins, $f_{UVA\;Mel}$, quantified in (**f**). In the example in (**e**), the average *g* and *s* coordinates of all the pixels of the native DHICA eumelanin solution (the center of the pink cercle) were used as reference coordinates for the $f_{UVA\;Mel}$ fraction calculation. This relative fraction is calculated as the distance (e.g., green and orange brackets) of every experimental data point to the reference native melanin position. The raw data are expressed as bar plots with data overlap (°); the bar represents the mean, the dotted line the median, and the error bar the ±95% confidence interval of the mean. Statistically significant *p*-values: ** *p* ≤ 0.01, *** *p* ≤ 0.001. The colored brackets indicate the ES—effect size (dark green—very strong [2–Inf], light green—strong [1.5–2], and yellow—moderate [0.8–1.5]). Only *p*-values associated with moderate to very strong ES are shown. On the right panel of each graph, we only show the comparisons of melanins with no exposure condition.

**Figure 5 ijms-24-04517-f005:**
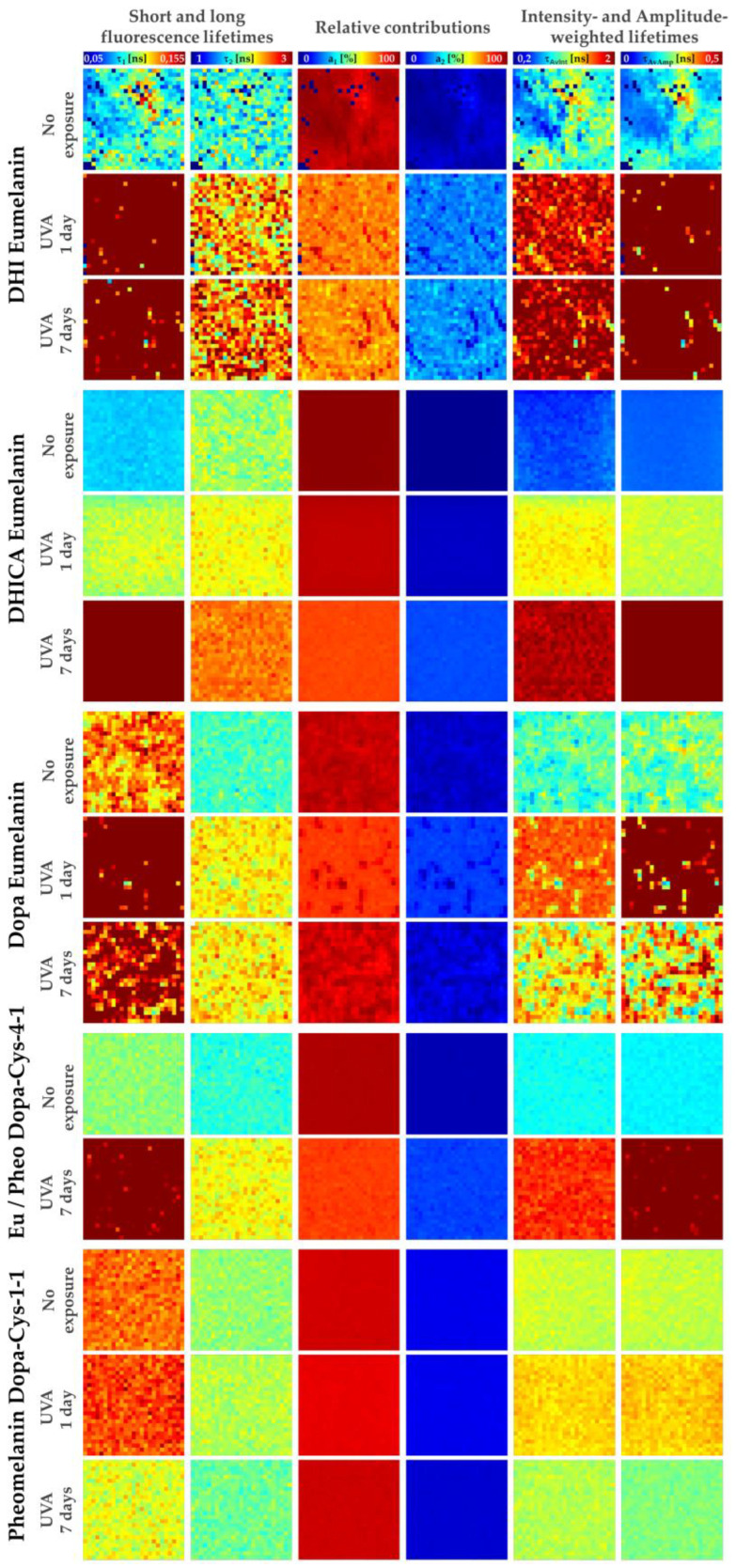
Multiphoton FLIM bi-exponential fitting analysis of native and UVA-exposed eumelanins, pheomelanin, and mixed eu-/pheo-melanins. For each condition, an example of calculated images (20 × 20 pixels spatial binning) is shown for all the parameters: $\tau_{1}$ short and $\tau_{2}$ long fluorescence lifetimes, their respective relative contributions, $a_{1}\left[\%\right]$ and $a_{2}\left[\%\right]$, as well as their combination parameters, the $\tau_{Av\;Int}$ intensity- and $\tau_{Av\;Amp}$ amplitude-weighted average lifetimes.

**Figure 6 ijms-24-04517-f006:**
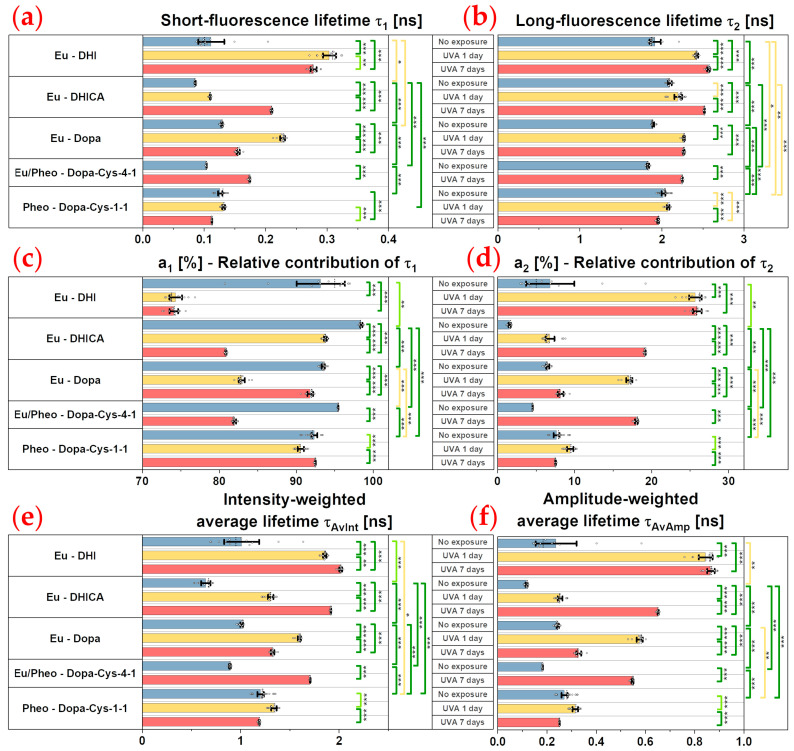
Multiphoton FLIM bi-exponential fitting analysis quantification results of native and UVA-exposed eumelanins, pheomelanin, and mixed eu-/pheo-melanins. For each type of melanin sample and exposure condition and every image pixel, we quantified the parameters of (**a**) $\tau_{1}$ short and (**b**) $\tau_{2}$ long fluorescence lifetimes and their respective relative contributions (**c**) $a_{1}\left[\%\right]$ and (**d**) $a_{2}\left[\%\right]$, as well as their combination parameters, the (**e**) $\tau_{AvInt}$ intensity- and (**f**) $\tau_{AvAmp}$ amplitude-weighted average lifetimes. The raw data are expressed as bar plots with data overlap (°), the bar represents the mean, the dotted line the median, and the error bar the ± 95% confidence interval of the mean. Statistically significant *p*-values: * *p* ≤ 0.05; ** *p* ≤ 0.01, *** *p* ≤ 0.001. The colored brackets indicate the ES—effect size (dark green—very strong [2–Inf], light green—strong [1.5–2], and yellow—moderate [0.8–1.5]). Only *p*-values associated with moderate to very strong ES are shown. On the right panel of each graph, we only show the comparisons of melanins with no exposure condition.

**Figure 7 ijms-24-04517-f007:**
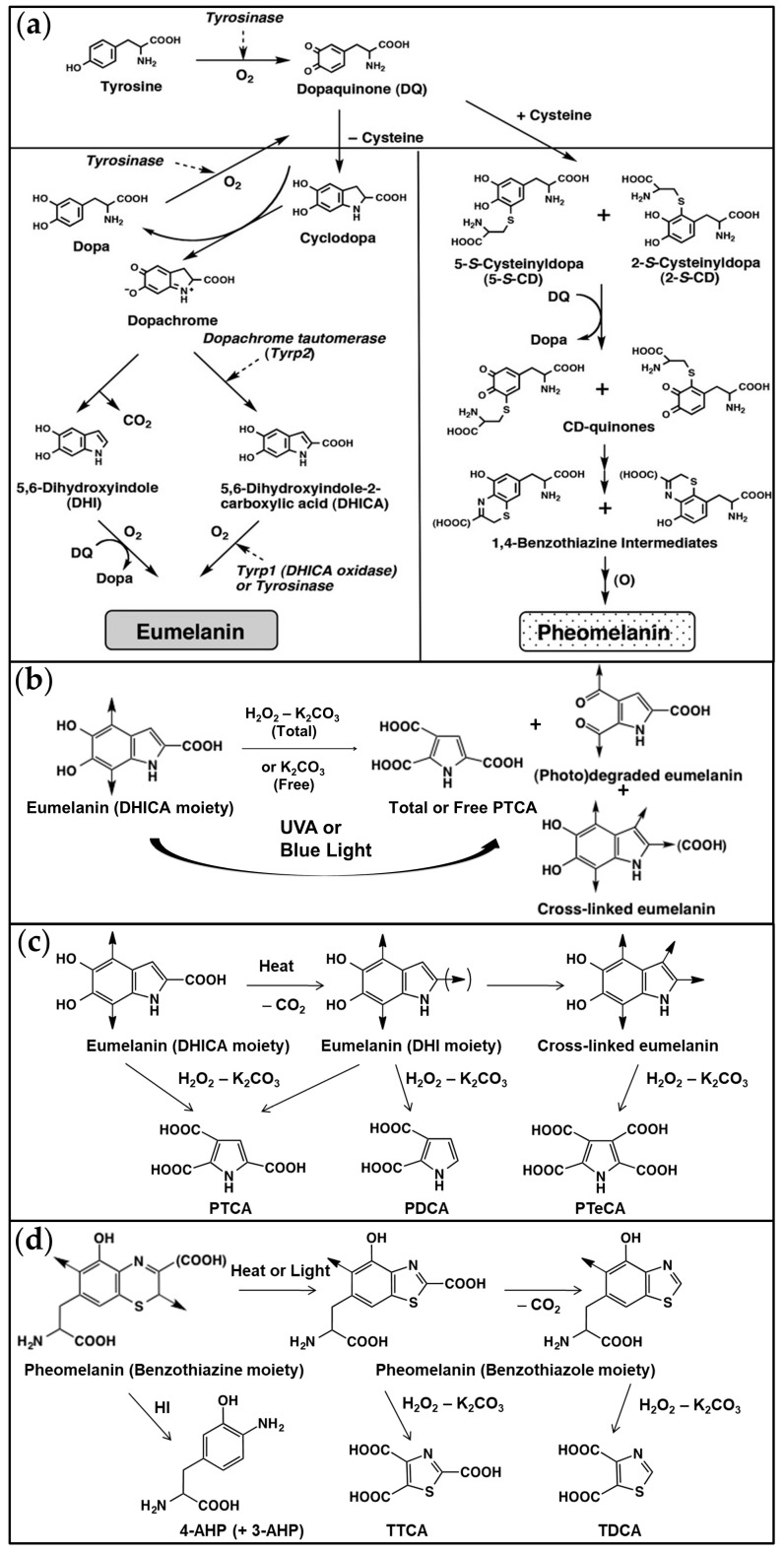
(**a**) Biosynthesis of eumelanin and pheomelanin. Tyrosinase, Tyrp1, and Tyrp2 are involved in eumelanin production, while tyrosinase and cysteine are required for pheomelanin production. Adapted with permission from ref. [68]. Copyright 2007, John Wiley and Sons. (**b**) Photo-induced structural modifications of eumelanin. Eumelanin consists of DHI and DHICA, but for the sake of simplicity, only DHICA is illustrated. DHICA gives PTCA after AHPO, while photo-induced oxidative degradation of PTCA gives free PTCA and photo-degraded eumelanin (diaryl ketone). The reaction of the DHI moiety with the indolequinone moiety gives crosslinked DHI. This structure gives PTeCA after AHPO. Adapted with permission from ref. [15]. Copyright 2018, John Wiley and Sons. (**c**) Effects of heat on structural features of eumelanin and their characterization by AHPO or reductive HI hydrolysis. Upon Hl hydrolysis of eumelanin, the DHICA moiety undergoes decarboxylation to form a crosslinked eumelanin via the DHI moiety. H_2_O_2_ oxidation of the DHICA moiety in eumelanin gives PTCA in a comparatively high yield while H_2_O_2_ oxidation of the DHI moiety gives PTCA and PDCA in low yields. H_2_O^2^ oxidation of crosslinked eumelanin gives PTeCA. Adapted with permission from ref. [45]. Copyright 2017, John Wiley and Sons. (**d**) Photo-induced structural modifications of pheomelanin and effects of light or heat on structural features of pheomelanin and its characterization by AHPO or reductive HI hydrolysis. Upon maturation of pheomelanin by heat or light, the benzothiazine moiety undergoes conversion into the benzothiazole moiety. Upon heating, the carboxylated benzothiazole moiety undergoes decarboxylation to form the decarboxylated benzothiazole moiety. HI hydrolysis of the benzothiazine moiety gives 4-AHP (and 3-AHP, not illustrated). H_2_O_2_ oxidation of the carboxylated and decarboxylated benzothiazole moieties gives TTCA and TDCA, respectively. Adapted with permission from ref. [45]. Copyright 2017, John Wiley and Sons.

**Figure 8 ijms-24-04517-f008:**
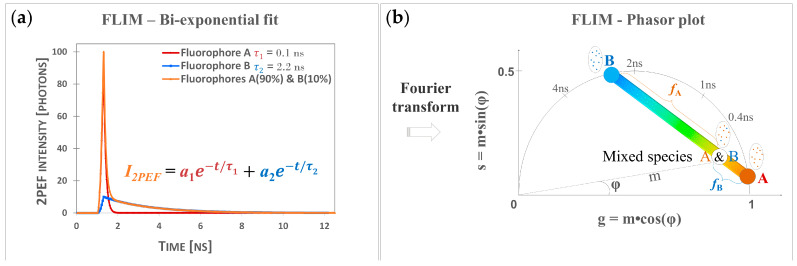
Principle of FLIM bi-exponential and phasor analyses. (**a**) Simulated mono-exponential (fluorophores A and B) and bi-exponential (mixed species of fluorophores A and B with, respectively, 90% and 10% relative contribution) two-photon excited fluorescence intensity decays (12.5 ns time range; 80 MHz); this shape of A&B bi-exponential decay can be measured in melanin-containing samples. In FLIM bi-exponential analysis, the 2PEF intensity decay is adjusted with the function in (**a**) to compute the values of short $\tau_{1}$ and long $\tau_{2}$ fluorescence lifetimes and their relative contributions $a_{1}\left[\%\right]$ and $a_{2}\left[\%\right]$. Images of FLIM bi-exponential fit and combination parameters such as amplitude- and intensity-weighted lifetimes are used for data analyses. (**b**) FLIM phasor analysis transforms a decay into a phasor with polar coordinates *g* and *s*, corresponding to the real and complex components of the Fourier transform, which can also be expressed as a function of *m* modulation and *φ* phase angle. Mono-exponential decays such as A and B will have their phasors on the semi-circle, whereas mixed species will have a phasor along a line connecting the two distinct lifetime phasors of A and B. The relative fractions $f_{A}$ and $f_{B}$ can be computed from the distances of A&B mixed species phasor to B and A phasors, respectively. Images of *g* and *s* as well as combination parameters such as the apparent phase and modulation lifetimes and their relative fractions are used for data analyses. Adapted from ref. [26].

**Table 1 ijms-24-04517-t001:** In tubo HPLC quantification results of native melanins (solutions and powders), melanin solutions exposed to UVA light at the radiance of 3.5 mW/cm^2^ for 1 and 7 days, and melanin powders heated at 100 °C for 8 days. Eumelanin and pheomelanin contents are indirectly estimated by HPLC analysis of specific degradation products obtained after AHPO, K_2_CO_3_ extraction, and HI hydrolysis (see Section 4.1 and Section 4.3). The data in µg/mg are given as the mean value estimated using 0.1 mL solution for each type of chemical degradation, performed on two separate occasions.

Melanin Type	Conditions	H_2_O_2_ Oxidation					K_2_CO_3_ Extraction	HI Hydrolysis		Ratio		
		Benzothiazole Pheomelanin	Benzothiazole Pheomelanin	DHI Eumelanin	DHICA Eumelanin	Eumelanin Crosslinking	DHICA Eumelanin Peroxidation	Benzothiazine Pheomelanin		DHICA Eumelanin Oxidation	Pheomelanin Oxidation	Eumelanin Crosslinking/ Photo-Degradation
		TDCA	TTCA	PDCA	PTCA	PTeCA	Free PTCA	4-AHP	3-AHP	Free/Total PTCA ×100	TTCA/ 4-AHP ×100	PTeCA/PTCA ×100
Eu - DHICA solution	No exposure	0.00	0.00	1.32	64.90	1.27	2.15	-	-	3.31	-	1.96
	UVA 1 day	-	-	-	46.10	-	6.87	-	-	14.90	-	-
	UVA 7 days	0.00	0.00	0.59	40.60	2.64	15.30	-	-	37.68	-	6.5
Eu - DHI suspension	No exposure	0.00	0.00	13.10	5.53	5.29	0.01	-	-	0.13	-	95.66
	UVA 1 day	0.00	0.00	12.40	5.81	6.14	0.01	-	-	0.22	-	105.68
	UVA 7 days	0.00	0.00	9.68	5.40	7.42	0.05	-	-	0.96	-	137.41
Eu - Dopa suspension	No exposure	0.00	0.00	11.50	5.36	3.76	0.01	-	-	0.21	-	70.15
	UVA 1 day	0.00	0.00	9.43	4.79	2.98	0.01	-	-	0.23	-	62.21
	UVA 7 days	0.00	0.00	9.36	5.65	4.18	0.03	-	-	0.50	-	73.98
Pheo - Dopa-Cys-1-1 solution	No exposure	3.37	7.51	1.30	4.64	0.36	0.06	133.50	26.00	1.29	5.63	7.76
	UVA 1 day	3.90	8.71	1.81	3.54	-	-	83.20	17.10	-	10.47	-
	UVA 7 days	6.20	14.00	2.41	2.75	0.36	0.14	39.50	11.30	5.09	35.44	13.09
Eu/Pheo - Dopa-Cys-4-1 suspension	No exposure	2.33	5.15	3.81	5.20	1.60	0.05	52.60	13.70	0.96	9.79	30.77
	UVA 7 days	3.08	7.58	3.42	4.10	1.64	0.05	21.60	6.45	1.22	35.09	40.00
Eu - DHICA powder	No exposure	0.00	0.00	2.52	123.00	1.00	-	-	-	-	-	0.81
	100 °C, 8 days	0.00	0.00	1.49	11.50	4.52	-	-	-	-	-	39.30
Eu - DHI powder	No exposure	0.00	0.00	3.04	2.32	3.82	-	-	-	-	-	164.66
	100 °C, 8 days	0.00	0.00	3.56	2.87	7.59	-	-	-	-	-	264.46
Eu - Dopa powder	No exposure	0.00	0.00	7.27	4.96	1.25	-	0.17	0.16	-	0.00	25.2
	100 °C, 8 days	0.00	0.00	6.87	6.89	2.03	-	0.14	0.32	-	0.00	29.46
Eu - DHI + DHICA 1-1 powder	No exposure	0.00	0.00	2.08	39.80	2.14	-	-	-	-	-	5.38
	100 °C, 8 days	0.00	0.00	1.95	8.21	4.26	-	-	-	-	-	51.89
Pheo - Dopa-Cys-1-1 powder	No exposure	2.30	2.78	10.30	3.83	0.78	-	133.00	27.70	-	2.09	20.37
	100 °C, 8 days	6.88	5.14	9.67	2.38	0.92	-	69.50	17.40	-	7.40	38.66
Eu/Pheo - Dopa-Cys-2-1 powder	No exposure	3.80	2.96	4.27	4.04	0.70	-	40.90	8.62	-	7.24	17.33
	100 °C, 8 days	8.78	6.05	3.27	2.37	0.90	-	15.50	4.36	-	39.03	37.97

**Table 2 ijms-24-04517-t002:** Mean values of multiphoton FLIM phasor and bi-exponential fitting quantification parameters of native and UVA-exposed melanin samples. For each type of condition, the data are given as mean values with standard deviation of the mean calculated over all the pixels of all the images.

Melanin Type	Conditions	$\tau_{1}$ $\left[ns\right]$	$a_{1}$ $\left[\%\right]$	$\tau_{2}$ $\left[ns\right]$	$a_{2}$ $\left[\%\right]$	$\tau_{AvInt}$ [ns]	$\tau_{AvAmp}$ [ns]	*g*	*s*	$\tau_{\varphi }$ $\left[ns\right]$	$\tau_{m}$ $\left[ns\right]$	$f_{UVA\;Mel}$
Eu - DHI suspension	No exposure	0.118 ± 0.052	91.78 ± 6.26	1.924 ± 0.299	8.22 ± 6.26	1.094 ± 0.374	0.270 ± 0.173	0.593 ± 0.115	0.154 ± 0.044	0.553 ± 0.219	2.611 ± 0.676	0.102 ± 0.070
	UVA 1 day	0.304 ± 0.067	74.28 ± 3.90	2.423 ± 0.257	25.72 ± 3.90	1.854 ± 0.183	0.846 ± 0.130	0.496 ± 0.039	0.289 ± 0.027	1.165 ± 0.139	2.843 ± 0.251	0.170 ± 0.028
	UVA 7 days	0.278 ± 0.076	74.10 ± 5.06	2.568 ± 0.370	25.91 ± 5.07	2.017 ± 0.270	0.866 ± 0.174	0.381 ± 0.044	0.239 ± 0.027	1.268 ± 0.200	3.957 ± 0.383	0.231 ± 0.041
Eu - DHICA solution	No exposure	0.085 ± 0.002	98.43 ± 0.27	2.096 ± 0.100	1.57 ± 0.27	0.649 ± 0.089	0.117 ± 0.007	0.888 ± 0.031	0.084 ± 0.021	0.190 ± 0.052	1.000 ± 0.167	0.032 ± 0.019
	UVA 1 day	0.109 ± 0.004	93.80 ± 0.55	2.202 ± 0.107	6.72 ± 1.18	1.302 ± 0.072	0.253 ± 0.022	0.677 ± 0.015	0.222 ± 0.010	0.653 ± 0.040	1.959 ± 0.070	0.252 ± 0.017
	UVA 7 days	0.210 ± 0.010	80.86 ± 0.53	2.517 ± 0.060	19.14 ± 0.53	1.916 ± 0.041	0.651 ± 0.022	0.589 ± 0.008	0.304 ± 0.006	1.026 ± 0.026	2.245 ± 0.045	0.371 ± 0.08
Eu - Dopa suspension	No exposure	0.129 ± 0.012	93.55 ± 1.47	1.893 ± 0.085	6.45 ± 1.47	1.007 ± 0.117	0.243 ± 0.038	0.817 ± 0.043	0.206 ± 0.033	0.507 ± 0.103	1.261 ± 0.184	0.044 ± 0.030
	UVA 1 day	0.228 ± 0.026	82.87 ± 2.41	2.269 ± 0.101	17.13 ± 2.41	1.597 ± 0.119	0.577 ± 0.082	0.661 ± 0.029	0.307 ± 0.022	0.926 ± 0.089	1.863 ± 0.116	0.186 ± 0.032
	UVA 7 days	0.155 ± 0.023	91.81 ± 2.62	2.267 ± 0.130	8.19 ± 2.62	1.326 ± 0.196	0.329 ± 0.083	0.694 ±0.065	0.227 ± 0.041	0.665 ± 0.170	1.854 ± 0.278	0.136 ± 0.055
Eu/Pheo - Dopa-Cys-4-1 suspension	No exposure	0.104 ± 0.003	95.45 ± 0.17	1.828 ± 0.057	4.55 ± 0.17	0.890 ± 0.031	0.182 ± 0.005	0.891 ± 0.010	0.172 ± 0.008	0.384 ± 0.019	0.917 ± 0.056	0.011 ± 0.006
	UVA 7 days	0.174 ± 0.013	81.96 ± 1.00	2.247 ± 0.076	18.04 ± 1.00	1.707 ± 0.053	0.548 ± 0.033	0.643 ± 0.014	0.318 ± 0.010	0.984 ± 0.042	1.928 ± 0.066	0.288 ± 0.014
Pheo - Dopa-Cys-1-1 solution	No exposure	0.126 ± 0.010	92.38 ± 1.00	2.018 ± 0.086	7.62 ± 1.00	1.200 ± 0.090	0.270 ± 0.032	0.777 ± 0.021	0.239 ± 0.018	0.614 ± 0.061	1.421 ± 0.086	0.025 ± 0.013
	UVA 1 day	0.132 ± 0.007	90.56 ± 0.82	2.076 ± 0.067	9.44 ± 0.82	1.340 ± 0.065	0.315 ± 0.024	0.733 ± 0.021	0.266 ± 0.010	0.723 ± 0.045	1.595 ± 0.091	0.053 ± 0.022
	UVA 7 days	0.113 ± 0.005	92.48 ± 0.30	1.953 ± 0.068	7.52 ± 0.30	1.189 ± 0.041	0.251 ± 0.010	0.769 ± 0.012	0.237 ± 0.009	0.613 ± 0.027	1.467 ± 0.059	0.015 ± 0.008

**Table 3 ijms-24-04517-t003:** List of synthetic melanins studied in their native, UVA oxidized, and crosslinked state using multiphoton FLIM microscopy and HPLC chemical analysis of melanin degradation products.

Melanin Type	Conditions	State
Eu - DHICA	No exposure	Solution (0.200 mg/mL, pH 6.8 buffer)
	UVA 1 day—3.5 mW/cm^2^	
	UVA 7 days—3.5 mW/cm^2^	
Eu - DHI	No exposure	Suspension (0.153 mg/mL, pH 6.8 buffer)
	UVA 1 day—3.5 mW/cm^2^	
	UVA 7 days—3.5 mW/cm^2^	
Eu - Dopa	No exposure	Suspension (0.153 mg/mL, pH 6.8 buffer)
	UVA 1 day—3.5 mW/cm^2^	
	UVA 7 days—3.5 mW/cm^2^	
Pheo - Dopa-Cys-1-1	No exposure	Solution (0.200 mg/mL, pH 6.8 buffer)
	UVA 1 day—3.5 mW/cm^2^	
	UVA 7 days—3.5 mW/cm^2^	
Eu/Pheo - Dopa-Cys-4-1	No exposure	Suspension (0.227 mg/mL, pH 6.8 buffer)
	UVA 7 days—3.5 mW/cm^2^	
Eu - DHI Monomer	No exposure	Powder (pale yellow), capped under argon
Eu - DHICA Monomer	No exposure	Powder (pale yellow)
Pheo - Benzothiazine Monomer	No exposure	Powder (pale blue), capped under argon
Pheo - Benzothiazole Monomer	No exposure	Powder (glassy)
Eu - DHICA	No exposure	Powder
	100 °C, 8 days	
Eu - DHI	No exposure	Powder
	100 °C, 8 days	
Eu - Dopa	No exposure	Powder
	100 °C, 8 days	
Eu - DHI + DHICA 1-1	No exposure	Powder
	100 °C, 8 days	
Pheo - Dopa-Cys-1-1	No exposure	Powder
	100 °C, 8 days	
Eu/Pheo - Dopa-Cys-2-1	No exposure	Powder
	100 °C, 8 days	

## Data Availability

The datasets generated during the current work are available from the corresponding author on reasonable request.

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
