# Peer review of "Multiphoton FLIM Analyses of Native and UVA-Modified Synthetic Melanins"

_ijms, 2023, doi:10.3390/ijms24054517_

Round 1

Reviewer 1 Report

Sir, 

I have reviewed the manuscript "Multiphoton FLIM analyses of native and UVA modified synthetic melanins " submitted by Ana-Maria Pena and co-workers to IJMS journal.

The topic is rather interesting and it can be clinically relevant in e.g. photoageing. Using various affordable F-techniques, one can easily address multiple  questions relevant to e.g. cutaneous biology. 

Sadly, The Introduction is poorly written. It is lengthy and requires very radical shortening. Obviously, it is important to postulate a hypothesis that the readers can follow. This manuscript is very technical, OK -  it is backed by a robust set of data. However, I would suggest also considering, e.g. biomedical applicability of these findings. Why do the authors not frame their story properly?  In lines 862-863 (!) we learn e.g. about the relevancy of their UV treatment selection (it is similar to solar radiance in Greece in June). This is exactly the 

In the abstract, the authors claim that their findings could be important for "in vivo human skin mixed melanins characterization". I believe that this aspect is very poorly addressed in the presented manuscript. By the way, there are recently no routine methods  (as far as I am aware of) available for these applications in the clinics (meaning for in vivo application/examination). Potentially, this would be even more tempting to address these melanin-related questions in the context of photo-ageing in e.g. biopsies (usually after fixation). 

I believe that the manuscript holds certain potential and I do not comment on technicalities. But it cannot be regarded as attractive for readers of a multidisciplinary journal as IJMS is.  I believe that it would be good to reconsider the structure of the presented manuscript (or completely change the audience and submit it to some highly focused journal). 

Minor points

Line 35: "manifestations of hyperpigmentation" .... for clinical dermatologists, this wording is inaccurate/misleading. Hyperpigmentations are clinical disorders. You mean increased pigmentation after UV exposure. Please, change the wording to avoid this confusion. 

Author Response

Dear reviewer 1,

We greatly thank you for all the comments that helped us improve our manuscript. Please find our answers to your questions and comments in the attached file.

Kind regards,

Ana-Maria Pena, on behalf of all co-authors

Reviewer 2 Report

The Authors of this manuscript have investigated the possibility of using  2-photon FLIM imaging, along with phasor and bi-exponential fitting, to perform chemical analysis of native and UVA exposed melanins. Despite the fact that this method has been extensively employed for the identification of melanins, the Reviewer recognizes the novelty of the study which is focused on the characterization of  DHI, DHICA eumelanins, benzothiazole/benzothiazine pheomelanins and mixed eu-/pheo-melanins, typically found in human skin and hair samples as well as their changes with UVA light exposure. The article is overall well-written, the materials and methods are well-described and the final conclusions are strongly supported by the results. I have only a few minor comments that could probably improve further the readability and the overall quality of the manuscript.

1.  Line 410: "...confidence interval of the mean and the  (······) the median line." Please correct the sentence if needed.

2. Line 637: "For this type of mixed melanins, the multiphoton FLIM and HPLC results are not in agreement." Could the Authors explain the reasons of disagreement between the two methods?

3.  Lines 693-695: The Authors are suggested to add a Reference at this point. 

Author Response

Dear reviewer 2,

We greatly thank you for all the comments that helped us improve our manuscript. Please find our answers to your questions and comments in the attached file.

Kind regards,

Ana-Maria Pena, on behalf of all co-authors

Round 2

Reviewer 1 Report

Sir, 

I have reviewed the updated version of the submitted manuscript "Multiphoton FLIM analyses of native and UVA modified synthetic melanins" submitted b Ana-Maria Pena and co-workers

The rebuttal butter is, in an orderly manner, point-by-point addressing all my previously listed critical remarks. I am grateful for very significant changes in the Introduction. The text is now clarified, and it is scientifically concise for the benefit of interested readers. Importantly, the authors also more clearly highlighted the potential importance of their findings and also p applicability to other research areas. 

I believe that the article can be accepted for publication.